# Decarbonization in Shipping Industry: A Review of Research, Technology Development, and Innovation Proposals

**George Mallouppas * and Elias Ar. Yfantis**

Cyprus Marine and Maritime Institute, Larnaca 6023, Cyprus; elias.yfantis@cmmi.blue
* Correspondence: george.mallouppas@cmmi.blue; Tel.: +357-99307697

**Abstract:** This review paper examines the possible pathways and possible technologies available that will help the shipping sector achieve the International Maritime Organization's (IMO) deep decarbonization targets by 2050. There has been increased interest from important stakeholders regarding deep decarbonization, evidenced by market surveys conducted by Shell and Deloitte. However, deep decarbonization will require financial incentives and policies at an international and regional level given the maritime sector's ~3% contribution to green house gas (GHG) emissions. The review paper, based on research articles and grey literature, discusses technoeconomic problems and/or benefits for technologies that will help the shipping sector achieve the IMO's targets. The review presents a discussion on the recent literature regarding alternative fuels (nuclear, hydrogen, ammonia, methanol), renewable energy sources (biofuels, wind, solar), the maturity of technologies (fuel cells, internal combustion engines) as well as technical and operational strategies to reduce fuel consumption for new and existing ships (slow steaming, cleaning and coating, waste heat recovery, hull and propeller design). The IMO's 2050 targets will be achieved via radical technology shift together with the aid of social pressure, financial incentives, regulatory and legislative reforms at the local, regional and international level.

**Keywords:** decarbonization; hydrogen; ammonia; biofuels; fuel consumption; slow steaming

## 1. Introduction

The shipping sector is crucial for international trade (~80–90% of the global trade occurs through shipping) and hence vital to the world economy [1]. Due to the scale of the sector, shipping represents ~3% of the total global green house gas (GHG) emissions [2], therefore strict environmental regulations around $NO_x$, $SO_x$ and $CO_2$ emissions are set to cause major technological changes in the industry [1,3]. For example, Liquefied Natural Gas (LNG) can improve the performance, and on the other hand, with methane slip, the benefits are reduced [1]. Other fuels and/or other technologies such as biofuels, hydrogen, nuclear and carbon capture and storage (CCS) could all decarbonize the industry, but each have significant barriers regarding cost, resources and social acceptability [1]. In addition, fuel consumption can be improved by various efficiency improvements (such as hull design and cleaning, and propeller design, to name a few). It is obvious that numerous problems/issues must be tackled in order to achieve deep decarbonization of the shipping industry. Thus, there is "no single route and a multifaceted response is required" [1] from different sectors of the industry.

Moreover, demand for shipping is likely to grow over the next three decades [2]. Shell and Deloitte conducted a market survey in order to understand the current market trends [4]. In their survey, more than 90% of respondents from the shipping industry considered decarbonization an important or top priority for their organizations. Eighty percent of the respondents also noted that the importance of decarbonization has "increased considerably over the past 18 months" [4]. This, therefore, shows that the market and industry are considering decarbonization as part of their business strategy over the coming

decades, in line with the International Maritime Organization's ambition to reduce $CO_2$ emissions from shipping by at least 50% by 2050 compared with a 2008 baseline [5].

Deep decarbonization will require financial incentives and policies at an international and regional level given the maritime sector's ~3% contribution to GHG emissions [1,6]. Maritime emission and reduction measures are commonly divided into two main categories: technical (ship size, ship–port interface, etc.) and operational (lower-speeds, waste heat recovery, etc.) [7,8]. The International Transport Forum recognizes additional separate routes to achieve decarbonization, which is the use of alternative fuels (sustainable biofuels, hydrogen, ammonia), electrification of ships and wind assistance, albeit these could be argued to fall under the technical measures' category [8].

The scope of this review paper is to survey the literature related to the shipping industry, based on research articles and grey literature, and discuss the potential routes to achieve deep decarbonization by 2050, as set by the International Maritime Organization's (IMO) targets. Section 2 discusses alternative fuels that can be used in shipping. The section will discuss the relative advantages and disadvantages of alternative fuels. Section 3 discusses renewable energy sources that are available within shipping, such as wind, solar and biomass. Section 4 discusses the maturity of technologies currently available that can help the shipping industry achieve deep decarbonization. Various technologies will be presented, such as Internal Combustion Engines, fuel cells, batteries, supercapacitors and nuclear energy. Section 5 discusses the various $CO_2$ abatement options; in essence, strategies and techniques that can help reduce fuel consumption such as vessel and propeller design and waste heat recovery. Finally, the review will conclude with a summary and future recommendations.

## 2. Climate Change and the Shipping Industry

### 2.1. Climate Change

The Intergovernmental Panel on Climate Change (IPCC) has set a limit rise of 1.5 °C in global temperature, hence shipping will need to "go beyond operational and energy efficiency and deploy zero-emission fuels and propulsion technologies" [2]. This is because the expected lifetime of vessels is a 20–30-year period, hence the maritime sector will need to ensure that zero-emission vessels are fully operational on deep-sea trade routes at a commercial scale by 2030 [2]. In this way, large-scale deployment will be possible beyond the 2030s [2].

Size of Shipping Industry in Terms of Emissions

Balcombe et al. [1] have assessed data from various sources [9–12]. They obtained the $CO_2$ emissions from global shipping and compared it with global trade and relative share of $CO_2$ emissions that originate from shipping. They show that $CO_2$ emissions have continuously increased since 1990, hand in hand with global trade. On the other hand, the share of $CO_2$ emissions decreased between 2007 to 2014 due to emissions from other sectors outside the shipping sector [1].

In 2014, international shipping emitted 1130 Mt $CO_2$, which accounts for 3.1% of global $CO_2$ emissions [1,11]. International Transport Forum [8] present the for the 2015 $CO_2$ emissions and 2035 baseline predictions, which is based on the freight model for the global shipping routes. They show that the main $CO_2$ emissions are produced along the East–West trade lines [8], and therefore emissions are expected to grow on these lines.

The greatest source of GHG emissions within shipping come from container ships, bulk carriers and oil tankers as opposed to domestic routes [1,11]. It is worthwhile mentioning that the IMO targets refer mainly to international shipping, although these targets have the potential to be enforced at a regional level. The IMO's targets are normalized over transport work, which, from this point of view, international shipping has lower $CO_2$ emissions. Balcombe et al. [1] processed data from Reference [13] and show that the $CO_2$ emissions' contribution from tankers can be distinguished from container ships (container ships are a

smaller fleet but with the highest emissions as opposed to tankers), due to their different operational profiles.

*2.2. Current Legislation and Incentives to Meet 2050 Emission Targets Set by the International Maritime Organization (IMO)*

The IMO is a United Nations (UN) agency, and its mission statement is to "promote safe, secure, environmentally sound, efficient and sustainable shipping through cooperation" [14]. In April 2018, the IMO defined the Initial Strategy with the objective to reduce GHG emissions from shipping by at least 50% by 2050 compared with a 2008 baseline [5].

The key ambitions of the Initial Strategy are [15]:

- To reduce the carbon intensity of international shipping, compared to 2008 levels, by 40% by 2030.
- To increase that reduction to 70% by 2050.
- To reduce the GHG emissions from international shipping, again compared to 2008 levels, by at least 50%, by 2050.
- To achieve zero GHG emissions as soon as possible within this century, i.e., by 2100.

The IMO will try to achieve these ambitions by including pre-existing energy efficiency measures as well as new measures as applicable in the short-, mid- and long-term [15]. Figure 1 shows the IMO's plan for ship improvements from 2013 to 2050 [16].

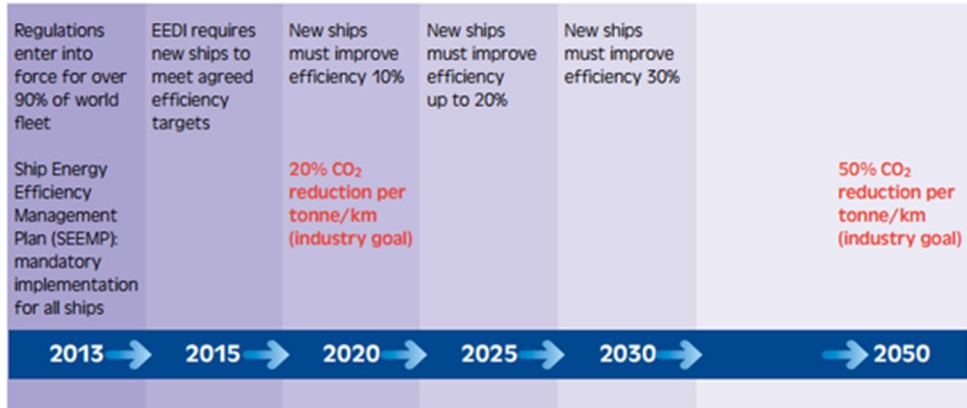

**Figure 1.** Diagram showing the International Maritime Organization's (IMO) plan for ship improvements from 2013 to 2050 (image obtained from Reference [16]).

Bouman, Lindstad, Rialland and Strømman discuss six main groups for maximum possible $CO_2$ emission reductions [17]. Namely, these are:

1. Hull design,
2. Economy of scale,
3. Power and propulsion (including energy-saving devices),
4. Speed,
5. Fuels and alternative energy sources, and
6. Weather routing and scheduling

Bouman, Lindstad, Rialland and Strømman [17] show the predicted annual $CO_2$ emissions until 2050 for business-as-usual (BAU) and the aforementioned reduction scenarios. From their study it is clear that to reach almost net-zero $CO_2$ emissions in the shipping industry, all six reduction scenarios will need to be implemented. In addition, Bouman, Lindstad, Rialland and Strømman mention of "an overlap between BAU and reduction scenarios in 2020 and in 2030, reflecting uncertainty in the scenarios as well as different assumptions on the rate of adoption of emission mitigation measures and growth rates of global maritime transport" [17].

### 2.2.1. Ship Energy Efficiency Management Plan (SEEMP)

The Ship Energy Efficiency Management Plan (SEEMP) is a management plan that improves fuel efficiency via operational improvements, and it is applicable to new and existing ships [1,15]. SEEMP for existing ships is a mandatory requirement [15]. The improvements offered by SEEMP are an optimized vessel speed, increased frequency of hull or propeller cleaning, or even by having different route choices to reach a destination (which includes avoiding heavy weather). It should be mentioned that SEEMP is specific to a ship because it takes into account unique factors, such as cargo, routes, dry docking schedule, as well as broader corporate or fleet level strategies [15].

### 2.2.2. Energy Efficiency Design Index (EEDI)

In 2001, Energy Efficiency Design Index (EEDI) was added to MARPOL [1]. The EEDI is applicable to new ships and it is a "monitoring tool which ship owners and operators can consult to gauge the potential impact of any management changes they make and thus weigh up the options from a more informed position" [15]. Hence, the EEDI via technical efficiency improvements reduces the $CO_2$ emissions [1]. The EEDI is the first global regulation to determine $CO_2$ emissions standards [18]. Depending on year of implementation, the International Council of Clean Transportation (ICCT) estimates that not all ships (globally) will be fully compliant with the EEDI regulations by the period 2040–2050 [18].

### 2.2.3. Energy Efficiency Operational Index (EEOI)

The Energy Efficiency Operational Indicator (EEOI) is an index proposed by the IMO to be used on a voluntary basis, intended to measure the efficiency of existing ships. The EEOI is defined as the ratio of mass of $CO_2$ emitted per unit of transport work ("capacity mile"). The index takes into account the fuel consumption (and the corresponding $CO_2$ emissions) as all fuel consumed at sea and in port during the time period, which refers to, by main and auxiliary engines, including boilers and incinerators [19].

### 2.2.4. New Short-Term Measures (2018–2023)

The IMO has developed new short-term measures (the Energy Efficiency Existing Ship Index (EEXI), which is promoted by Japan to meet the IMO 2030 targets [20], and the Carbon Intensity Index (CII)) [21]. These measures were an outcome of the ISWG-GHG 7 meeting and effectively developed new measures from different groups of countries under the two categories:

(1) Technical: For existing ships EEXI, taking EEDI and applying to existing ships [22].
(2) Operational: Addition of a mandatory Carbon Intensity Index with a rating scheme from A to E [22]. The measure is applied to all existing ships with a certain size threshold.

It should be mentioned that the measures are a consolidated balance to achieve a political compromise [22].

### 2.2.5. Energy Efficiency Existing Ship Index (EEXI)

In November 2020, the IMO approved amendments to MARPOL Annex VI, which introduced a new measure for existing ships, namely the Energy Efficiency Existing Ship Index (EEXI) [23]. The EEXI will be enforced by 2023 and "will be applicable for all vessels above 400 GT falling under MARPOL Annex VI" [23]. Effectively, the EEXI is considered to be an extension of the EEDI, and in essence, the "required EEXI is almost in agreement with requirements" for new build ships [23].

The EEXI describes the $CO_2$ emissions per cargo ton and mile and "determines the standardized $CO_2$ emissions related to installed engine power, transport capacity and ship speed" [23]. In other words, the EEXI limits the amount of $CO_2$ emitted per unit of transport supply [24]. Note that the EEXI is a technical (or design) index rather than an operational index. Therefore, there are no measured values of previous years and

no onboard measurements are required. In essence, the EEXI only refers to the ship's design [23].

Except for existing ships built in accordance with the EEDI Phase 2 or 3 requirements, a technical file that includes a technical calculation of the attained EEXI of a given ship must show that it is lower than the required EEXI value [23].

Rutherford, Mao and Comer [24] mention that, as proposed, the EEXI will only make a minor contribution to the $CO_2$ emissions reduction targets set by the IMO. Rutherford, Mao and Comer [24] specifically mention that, compared to the baseline without EEXI, the EEXI would only contribute 0.7–1.3% $CO_2$ emissions reductions from the 2030 fleet due to the continuing dominance of slow steaming (vessel speed of 20–22 knots (39 km/h)).

### 2.2.6. Carbon Intensity Index (CII)

There are proposals for ships greater than 5000 GT to submit their required annual operational carbon intensity indicator (CII) [21]. "The CII determines the annual reduction factor needed to ensure continuous improvement of the ship's operational carbon intensity within a specific rating level", with a rating scheme from A to E [21,22]. As opposed to EEXI, the CII is an operational short-term measure. The CII performance will be recorded within the ship's SEEMP [21].

### 2.2.7. Medium- (2023–2030) and Long-Term Measures (Beyond 2030)

Medium- and long-term measures will further develop the short-term measures. The measures will also take into account the implementation of market-based-mechanisms and the provision of incentives for the reduction of emissions. This implies synergies of technical, political and infrastructural measures in order to achieve the goal of GHG reductions by 2050 [1].

### 2.2.8. Market-Based Measures (MBM)

In view of the projections predicting growth of the shipping industry, it is believed that operational and technical measures are not sufficient to satisfactorily reduce GHG emissions, and hence there is a general consensus that market-based measures (MBM) as part of a general comprehensive package of measures will help achieve the IMO's targets [25]. MBM, discussed in the Marine Environment Protection Committee (MEPC), are expected to be over the medium term [26]. However, just like $CO_2$ emission reduction measures, the discussions have been hampered due to difference in opinions by different stakeholders [26,27]. MBM discussions begun at MEPC 56 in 2006 but have not progressed since MEPC 65 in 2013 [28].

MBM measures are based on economic variables and/or tax levies and serve two main purposes [25,29]:

1. Economic incentives for the maritime sector to reduce its fuel consumption by investing in more fuel-efficient ships, technologies and operating ships in a more energy efficient way.
2. Offsetting in other sectors with growing emissions related to the shipping industry.

There are several proposed MBMs schemes, but none have come into fruition: examples are contribution of $CO_2$ emissions where they are transferred to funds via emission trading systems and schemes based on a ship's efficiency based on design and operation [25].

Possibly, MBM will have a dominant role on IMO's strategy in the near future. However, as Jorgensen [30] correctly mentions, MBM must be governed by "global rules" in order to avoid paying a levy on $CO_2$ emissions multiple times. Another issue with MBMs is that the available models in the literature are based on the short-term period (i.e., are not based on an optimal control approach) [28]. Therefore, it is inherently difficult for a shipping company to estimate the reduction of $CO_2$ emissions, since the available models cannot analyze how much investment in capacity and how much effort is required to improve fuel efficiency [28].

### 2.3. European Green Deal

The discussion of MBMs at the IMO seems to have been put on hold, however the EU via the European Green Deal seems to be moving in the direction of MBMs [29]. The EU wants to be carbon-neutral by 2050, which includes shipping (although for shipping this is 90%), and by 2030, a 55–60% reduction of $CO_2$ with the 1990 baseline [22]. On the other hand, the IMO wants to cut total GHG emissions from international shipping by 50% by 2050 compared to the 2008 baseline, irrespective of maritime trade growth [31].

There is a strong push from the European Commission (EC) and European Parliament (EP) to include shipping in European Trading System (ETS), which is a carbon trading system. Currently, the timing and scope to include shipping in ETS is under debate [22]. The European Climate Law, which will be in effect by mid-2021, will be a key driving force for further actions including shipping in ETS [22]. Moreover, ETS will progress under the European Climate Law and the zero pollution action plan for water, air and soil, which will be in effect in 2021 [22]. ETS will act as a "regulatory tool" to reduce emissions from shipping [29].

The European Community Shipowner's Association [31] published a position paper regarding the European Green Deal related to shipping. They rightfully mention that emissions cannot be attributed to individual nations as it involves international transportation and so expects EU initiatives to contribute to the derivation of global rules, which of course will need to be accepted by all parties. Therefore, this would improve the competitive position of the EU shipping industry [31].

### 2.4. International Maritime Research and Development Board (IMRB)

To achieve such colossal reductions in emissions, as set by the Paris Agreement and the Initial Strategy of the IMO in the shipping sector, there are proposals from the industry to accelerate the development of low-carbon and zero-carbon emission technologies [22]. As a result, the International Maritime Research and Development Board (IMRB) was established. The IMRB is funded by a mandatory payment per ton of fuel oil purchased, which will generate a ~$5 Billion fund over the life of the program. This program is likely to generate market-based measures but has not been discussed or no general consensus has been reached by the IMO [22].

### 2.5. Existing Pilot Projects

Currently, there are 66 zero-emission pilot and demonstration projects for shipping globally under the Global Maritime Forum, many involving hydrogen fuels [32]. The uptake of pilot and demonstration projects is crucial in promoting the transition to zero emission fuels for the maritime industry. In order to achieve this, innovating pilot projects can help stimulate the uptake of zero-emission technologies by utilizing expertise from research institutions and industry. The aim of these pilot projects is to show the viability of the technologies to achieve zero emissions via [32]:

1. Safety,
2. Environmental impact,
3. Economic costs, and
4. Generation of knowledge and experience that can be used to further improve technologies. In this way, the overall cost will reduce, a key prerequisite for disseminating and scaling these technologies.

Japan is also working on four pilot projects, which are based on its report on "Roadmap to Zero Emission from International Shipping" [20]. These pilot projects are based on four alternative concepts to achieve zero-emissions [20]:

1. Hydrogen-fueled ship (C—ZERO Japan $H_2$).
2. Super-efficient LNG-fueled ship (C—ZERO Japan LNG and Wind).
3. Ammonia-fueled ship (C—ZERO Japan $NH_3$).
4. Onboard $CO_2$-capturing ships (C—ZERO Japan Capture).

## 3. Alternative Fuels for Shipping

Compared to other sectors such as road transportation and aviation, the shipping sector uses less refined and/or processed fuels [33]. The primary fuel used in the shipping sector to power marine diesel engines is heavy fuel oil (HFO). HFO has very high viscosity and contains large amounts of sulfur [33], which when combusted releases harmful $SO_x$ emissions. The shipping sector uses other fuels with lower viscosity levels and lower sulfur content, such as the marine gas oil (MGO) and marine diesel oil (MDO), the former being used for smaller vessels [33].

Alternative fuels may have the potential to lower or have zero net emissions when used for ship propulsion [8]. Currently, the use of alternative fuels, such as LNG and hydrogen, are gaining traction [34]. Alternative fuels can also be used as "drop-in" fuels (such as biodiesel) but are still applied in shipping on an experimental basis [34]. Table 1 lists possible $CO_2$ emissions reductions from alternative fuels and energy sources ([8], p. 32), which are further discussed in this section. Note that Japan is working on four pilot projects (identified in Section 1), which are based on its report on "Roadmap to Zero Emission from International Shipping" [20]. The report mentions that pilot concepts are based on two possible emission reduction pathways: (1) LNG, provided that LNG transitions to carbon-recycled methane, and (2) adoption of hydrogen/ammonia as fuel [20].

**Table 1.** Possible $CO_2$ emission reductions via the usage of alternative fuels (data obtained from Reference [8]). Note that 100% is achieved only if renewable energy is involved taking into account upstream emissions and electricity.

| Measures | Possible $CO_2$ Emissions Reductions |
| --- | --- |
| Advanced biofuels | 25–100% |
| Liquefied Natural Gas (LNG) | 0–20% |
| Hydrogen | 0–100% |
| Ammonia | 0–100% |
| Fuel cells | 2–20% |
| Electricity | 0–100% |
| Wind | 1–32% |
| Solar | 0–12% |
| Nuclear | 0–100% |

DNV GL [35] illustrate the various energy densities for different fuels. The energy density partly illustrates "how applicable the fuel is for certain ship types and ship operations" [35]. DNV GL [35] show that LNG has around "40% lower volumetric energy density than diesel, roughly the same as LPG" [35]. However, when the storage system is taken into account, LNG has roughly a third of the volumetric energy density as diesel. DNV GL [35] also mention that for liquid hydrogen, ammonia and methanol, their volumetric energy density is even lower, ~40–50% of LNG. Note that biodiesel "is the only fuel which is close to matching the energy density of diesel" [35].

In the same study DNV GL [35] compare the fuel cost/technology pathways. The main takeaways are [35]:

- LNG, methanol and LPG are competitive in terms of energy costs, while HVO is significantly more expensive.
- Hydrogen and ammonia are also far more expensive.
- The large cost range indicates a significant uncertainty in terms of pricing.

Lloyd's Register and University of Maritime Advisory Services (UMAS) [36] performed a technoeconomic study of carbon fuels with the potential of becoming a zero-carbon emitter. Lloyd's Register and UMAS [36] concluded that in the short term, biofuels look marginally more competitive than fuels derived from renewable electricity or from natural gas with carbon capture and storage. However, biofuels have challenges in terms of sustainability and availability, which, thus, in the mid–long-term, "any biofuel pathway is uncompetitive and prone to restrictions or higher prices" [36]. Therefore, it does not

necessarily make biofuels more competitive than hydrogen or ammonia that are derived from natural gas or renewable energy [36]. Lloyd's Register and UMAS also mention that, although some fuel pathways may be more resilient than others, fuel price is the dominant factor that determines the total cost of operation (TCO), therefore a fuel derived from natural gas or from a renewable energy source "may offer longer term benefits" as opposed to biofuel considering the future growth in energy global demand and the aforementioned sustainability and availability issues of biofuels [36].

From a technology readiness level (TRL) viewpoint, methanol, LNG and diesel are more mature compared to hydrogen and ammonia as "rules and regulations currently exist and there are vessels already using these fuels" [36]. There is also little difference for onboard technologies, "for example, between using bio-methanol, e-methanol or NG-methanol; the same applies to LNG (bio-LNG, fossil-LNG and e-LNG)" [36].

### 3.1. Hydrogen

Hydrogen has the advantage that no $CO_2$, particulate matter (PM) or $SO_x$ are released while burning, however, its availability and low volumetric energy density require significant additional infrastructure and system design [37]. Although, $NO_x$ emissions can be generated at significant levels, if the flame temperatures are greater than 1700 K and air is the oxidant. Nearly all hydrogen is produced from fossil fuels, in fact 6% of $CH_4$ and 2% of coal are used to generate hydrogen [38]. Hydrogen can be used as a fuel in fuel cells or as a blend in existing conventional diesel fuels commonly used in the shipping industry, such as heavy fuel oil (HFO) [8]. However, burning hydrogen as a "drop-in" fuel in marine diesel engines is possible at low levels of blending without significant risks of engine damage [39].

The IEA show that the global demand for pure hydrogen for the period 1975–2018 has increased [40], and this illustrates the fact that demand for hydrogen is growing, hence the current "hype" for hydrogen is here to stay for the long run. The cost of producing hydrogen is the greatest challenge, especially for green hydrogen (hydrogen produced from renewable energy): "A study by the Hydrogen Council determined that, using an import price of $3 per kilogram of hydrogen to power turbines, the electricity produced would cost about $140/MWh. By comparison, a 2019 estimate of electricity's levelized cost suggests the unsubsidized cost of natural gas combined-cycle electricity generation is between $44/MWh and $68/MWh" [41]. Furthermore, hydrogen does not have a standardized design and fueling procedure for ships and its bunkering infrastructure [42]. Currently, there are two techniques that are used to produce hydrogen: by steam methane reforming and water electrolysis [43].

Hydrogen production, storage, delivery and utilization are important issues for a sustainable Hydrogen Economy [44]. In addition, to achieve wider acceptance for hydrogen as a viable fuel, safety and reliability issues must be addressed, bearing in mind that public opinion may have a direct impact on policies [44]. Therefore, hydrogen safety, due to its volatility, is a major issue to be considered [8]. Hydrogen has high capital cost of storage, and onboard a ship there is a loss of cargo-carrying capacity [36]. Therefore, significant improvements are needed to first reduce capital cost of storage and secondly to resolve the onboard storage issues which are necessary to improve hydrogen's competitiveness compared to other alternative fuels [36].

#### 3.1.1. Hydrogen Storage Technologies

Hydrogen storage applications can be divided into two main categories: (1) stationary (for on-site storage and stationary power generation), and (2) mobile (transportation of hydrogen to final destination) [44]. In this review paper, the discussion will focus on hydrogen storage technologies applicable to the shipping industry, because maritime applications have different challenges compared to stationary or automotive applications [45]. Hydrogen under standard conditions has a low volumetric energy density, hence the need for efficient storage, i.e., hydrogen storage requires (1) high pressures, and/or (2)

low temperatures and/or (3) material that can store hydrogen in physical carriers (via adsorption) or chemical carriers [44,45].

Van Hoecke et al. [45] present the potential applicability of hydrogen storage methods for the shipping industry, namely, these are:

1.  Compressed hydrogen,
2.  Liquid hydrogen (cooling hydrogen to $-252.9\ °C$),
3.  Chemical storage

    a.  Ammonia, nitrogen-based storage (via the Haber-Bosch reaction, ammonia can serve as storage medium and a fuel itself),
    b.  $CO_2$-based storage (via carbon capture and storage, producing synthetic fuels such as synthetic diesel, synthetic liquefied synthetic methane—LSM, methanol and formic acid),
    c.  Aromatic Liquid Organic Hydrogen Carriers (LOHCs) (use of aromatic LOHCs and a catalyst to store hydrogen)

4.  Metal hybrides (store hydrogen within materials).

Compressed hydrogen is currently the most accepted and used storage method in the shipping industry. Such system is being used onboard ships, such as the FCS Alsterwasser inland passenger ship, which has been sailing since 2008 [46,47]. Liquid hydrogen has several complexities and currently there is no large-scale demand for the shipping industry (apart from a pilot project by Kawasaki Heavy Industries discussed later). Ammonia, nitrogen-based storage and $CO_2$-based storage are used in other industries and hence have lots of potential for knowledge transfer. Metal hybrides is quite promising, but still, further research and development is required to reach commercialization, but they have been used in submarine applications [47,48].

### 3.1.2. Hydrogen Safety

Hydrogen is a small molecule and can thus easily leak, especially when stored in a compressd state [45]. Due to its low ignition energy, it can lead to explosions which can be detrimental on a ship. The effects of low ignition energy are discussed later when hydrogen is used in internal combustion engines leading to knock events. In general, storage tanks for compressed hydrogen can withstand high pressures and have generally high safety factors (for the interested reader, Moradi and Groth [44] and van Hoecke et al. [45] discuss the different types of compressed storage tanks—Type I–IV). Materials under cryogenic conditions must withstand extremely low temperatures ($-252.9\ °C$) otherwise they become brittle. In addition to the risk of hydrogen explosion, spills under cryogenic conditions onboard are particularly hazardous because they can damage the hull of a ship by cold fracture [45,49]. Storage via chemical carriers poses the risk due to the handling of toxic materials (depending on the chemical storage method) such as ammonia and formic acid [45].

### 3.1.3. Green Hydrogen

Given the large scale of the shipping industry, which emits nearly 3% of all GHG emissions, green hydrogen may be one of the ways forward [50]. Green hydrogen is produced from renewable sources such as biomass, solar and wind through the process of electrolysis. It is worthwhile mentioning that the costs to produce green hydrogen are projected to fall significantly in the next decade due to economies of scale, technological improvements as well as renewable deployment. These could make green hydrogen price-competitive compared to grey and blue hydrogen (other colors for hydrogen are defined in the next section) [51]. Note that costs for producing green hydrogen have dropped by 50% since 2015 and could be further reduced by 30% by 2025 due to the benefits of increased scale and standardized manufacturing, among other factors. Currently, green hydrogen costs €0.1–0.15/kWh, whereas grey hydrogen (produced by reforming methane from natural gas, which releases $CO_2$ in the atmosphere) costs €0.045/kWh [38]. However,

green hydrogen is "already cost competitive" in niche applications, provided that recent market trends continue, and current policy mechanisms are maintained [52].

### 3.1.4. Other "Colors" for Hydrogen (Brown, Grey, Yellow, Blue)

Yellow hydrogen is produced from nuclear power, but unlikely to be used due to "political reasons". Port authorities will be unwilling to accept foreign ships with a nuclear reactor onboard [53]. Grey hydrogen is produced from methane reformation, autothermal reformation, partial oxidation, low-temperature plasma reformation and reformer, electrolyzer and purifier, whereas brown hydrogen is produced from coal. These emit GHG emissions. Blue hydrogen is produced from natural gas, but $CO_2$ emissions are captured (CCS). The ideal color is green, whereby hydrogen is produced from renewable energy sources (wind, solar, hydropower, etc.).

### 3.1.5. Liquid Hydrogen ($LH_2$)

According to Shell [54], liquid hydrogen ($LH_2$) has many advantages compared to other potential zero-emission fuels for shipping, therefore giving a higher likelihood of success. On the other hand, storage of $LH_2$ is complicated and costly, with a large number of safety issues due to the requirement of cryogenic storage (i.e., at high pressures and low temperatures: $-252.9\,°C$). An additional major issue with $LH_2$ is the fueling process that occurs at low temperatures. Specific insulation materials are required for the tank materials in order to avoid evaporation of $LH_2$ and hence avoid large heat fluxes into the tank [45,55]. Studies on novel insulation systems under cryogenic conditions, such as the one proposed Zheng et al. [55], can bring benefits to the shipping industry if $LH_2$ is the way forward.

Apart from the first pilot project by Kawasaki Heavy Industries transporting $LH_2$ in a tanker ship [45,56], no large-scale shipping exists for liquid hydrogen because of the storage complexities and the unavailability of a global market. In the pilot study by Kawasaki Heavy Industries [56], it was found that it is technically and economically possible to transport and store $LH_2$ from Australia to Japan.

### 3.2. Ammonia

Ammonia is produced commercially via the Haber-Bosch process, which combines hydrogen and nitrogen with the help of high temperatures and a catalyst [8]. Green ammonia can be produced by employing renewable energy sources such as solar, wind or hydropower, "which gives ammonia a comparative advantage compared to the production of HFO" [8]. However, producing green ammonia is not yet cost-competitive compared to conventional ammonia, where "90% of its production relies on fossil fuels such as natural gas" [8]. There are several global initiatives to produce green ammonia, such as the ammonia producer Yara (one of the world's leading ammonia producers) is planning to build a demonstration plant for the next-generation green ammonia synthesis plant that will utilize solar energy by 2025 [57,58].

Ammonia, compared to hydrogen, allows more hydrogen storage in liquid form without the need to use cryogenic storage ($-33.4\,°C$ for ammonia compared to $-252.9\,°C$ for hydrogen), thus making $NH_3$ a suitable hydrogen carrier [8]. The capital cost required to store hydrogen is far more expensive compared to ammonia, despite the fact that the energy density of both fuels is similar [8]. Similar to hydrogen, ammonia can be used as drop-in fuel in diesel engines, gas turbines and as a primary fuel in fuel cells, thus making ammonia a very attractive and competitive option [8]. Regarding the latest developments and ongoing projects for ammonia in Internal Combustion Engines (ICE) and fuel cell applications, the reader is directed to Section 5.

### 3.3. Methanol

Methanol may have a potential as it offers simpler handling and lower investment costs [59]. Methanol is an attractive fuel because it is low in carbon and sulfur-free. The methanol international market is going through a phase of huge expansion [60]. Methanol

can be produced from methane and it could serve as a possible future marine fuel [8]. Methanol can offer ~25% $CO_2$ emissions reduction potential compared to HFO [8]. In addition, methanol can reduce $SO_x$, $NO_x$ and PM by 99%, 60% and 95% respectively, "however, methanol can also be produced from renewable energy resources, such as $CO_2$ capture, industrial waste, municipal waste or biomass", in this way, the GHG effect can be significantly reduced [8].

Methanol has a relatively low flash point, is toxic and its vapor is denser than air [61]. The risk and safety analysis in the SPIRETH project have contributed to the development of ship classification society rules for methanol as a ship fuel. The work has also contributed to IMO's draft IGF code and class rules [62]. Methanol is plentiful, available globally, readily miscible in water, biodegradable and it can be 100% renewable. The life-cycle environmental footprint of bio-methanol is "greener" compared to LNG [63].

The Effship project evaluated different technical solutions and marine fuels that will be able to satisfy the $SO_x$ and $NO_x$ reductions regulations "in the short term (2015–2016), GHG reduction targets in the medium term (2030) and long term". It was concluded that methanol was the "best alternative fuel, taking into account prompt availability, use of existing infrastructure, price and simplicity of engine design and ship technology" [64,65].

The first ever methanol-powered ship is Stena Germanica, a project supported by the EU Motorways of the Seas program. Stena Germanica is a large passenger and car ferry. The project "converted a RoPax vessel into a methanol-powered vessel and provided bunkering as well as other necessary facilities in ports" [8].

### 3.4. Liquefied Natural Gas (LNG)

Liquefied Natural Gas (LNG), compared to its gaseous state, takes up 600 times less space for storage and transportation [66], hence natural gas is liquefied by cooling it to $-162$ °C. Currently, LNG is the cleanest available fuel for shipping which can be produced in meaningful volumes [54] and can comply with the $SO_x$ and $NO_x$ requirements while reducing $CO_2$ emissions (20–30%) [1]. In particular, LNG significantly reduces pollution from $NO_x$ and particulate matter (PM) compared to conventional marine fuels, while cutting emissions of $SO_x$ by more than 90% [1,54,67]. LNG is also a cost-effective fuel.

On the other hand, LNG can offer moderate environmental benefits due to methane slip (leaking of methane gas into the atmosphere), which varies across engine types [1]. Current LNG engines have a methane slip of 2–5% [68,69]. In this way, LNG may have short-term promise with minor policy intervention [1]. There are several lifecycle studies that analyze GHG emissions per kWh for LNG as shipping fuel. Balcombe et al. [1] have analyzed the various lifecycle studies [69–72] and estimated that the average total GHG emissions, which includes combustion and upstream, is 650 $gCO2_e$/kWh. Note that upstream denotes resource extraction, processing and liquefaction and transportation, and downstream emissions occur from combustion and methane slip.

There are four main technologies that use LNG as a fuel with different characteristics, efficiencies and emissions profiles [73]. The engines/turbines in use in the shipping sector today are [73]:

- Lean-burn spark ignition,
- Low-pressure dual fuel (4- and 2-stroke),
- High-pressure dual fuel and
- Gas turbine.

The AIDAnova is the world's first cruise ship powered by LNG. Shell LNG worked with the Carnival Corporation PLC to achieve reduced emissions, quieter engines and no visible emissions [54], and it is an example of a successful project powered by LNG. The AIDAnova has achieved a 95–100% reduction in $SO_x$, up to 85% reduction in PM and $NO_x$ and up to 22% tank to wake greenhouse gas emissions reduction compared to marine diesel oil [54]. Tom Strang, Senior Vice President Maritime Affairs at Carnival said: "LNG is currently the most cost-effective solution that offers emission reductions today. What we can achieve today with fossil LNG plus an increasing introduction in the percentage of

renewable liquefied methane—either from bio or synthetic sources—could see ships being effectively carbon-neutral in the future" [54].

Lindstadt et al. [74] quite notably mention "overall, in the public debate, LNG seems to be treated as a bipolar issue". They further mention that some studies argue that LNG is beneficial for both GHG and local air quality, however, other studies argue against LNG due to its high methane slip through the whole well-to-tank chain lifecycle analysis [74]. LNG contains carbon in its molecular formula, and hence it cannot decarbonize the shipping industry but can help the shipping industry to achieve decarbonization as a transition fuel [74]. Table 2 summarizes the advantages and disadvantages of LNG as a transition fuel for the shipping industry.

**Table 2.** Summary of advantages and disadvantages of LNG as a transition fuel for the shipping industry.

| Advantages | Disadvantages |
|---|---|
| No sulfur in its molecular formula, hence no $SO_x$ emissions | Slip methane (un-combusted methane leak) enhances GHG gains compared to traditional fuels such as MGO and HFO [74,75] |
| Lower $CO_2$ emissions because of C:H ratio (25% less compared to diesel or bunker fuel), so can help the industry to decarbonize as a transitional fuel [74] | Still an emitter of $CO_2$, hence cannot achieve decarbonization and need to take into account embodied $CO_2$ (well-to-tank analysis) |
| Compared to its gaseous state, takes up 600 times less space for storage and transportation [66] | But special care is required for transportation and storage |
| Compared to hydrogen and ammonia storage, lower implications in terms of toxicity and safety | Safety issues (explosion hazard or cold fracture of hull due to low temperatures) |
| Low $NO_x$ emissions if low-pressure dual fuel engine is used [74] | High $NO_x$ emissions if high-pressure dual fuel engine is used [74]. Depending on engine conditions, CO and unburnt hydrocarbons may increase [76]. |

## 4. Renewable Energy Sources

Renewable energy can either be used to generate green fuels or used directly for propulsion. "The development of renewable energy solutions for shipping has been hampered by over-supply of fossil fuel-powered shipping in recent years and the related depressed investment market" ([39], p. 4). The main barriers to increased penetration of renewable energy solutions for shipping remain: (1) the lack of commercial viability of such systems, and (2) limited motivation for deployment of clean energy solutions due to split incentives between ship owners and operators ([39], p. 4). Mofor, Nuttal & Newell ([39], pp. 10–11) tabulate a summary of various renewable energy applications, pathways, as well as their potential application for the shipping industry.

There are numerous potential renewable energy sources for shipping applications, namely: wind (soft sails, fixed wings, rotors, kites and conventional wind turbines), solar photovoltaics, biofuels, wave energy, batteries and supercapacitors, where the latter two are charged with renewable energy [39,77–83]. Renewable energy may be introduced in shipping via the following two routes: (1) as retrofits for the existing fleet, or (2) inclusion into new designs. Regarding new design concepts, and independent of ship size, most of the renewable energy will supply power for auxiliary and ancillary applications. However, very few designs will target 100% renewable energy or zero-emissions for primary propulsion (examples include B9, Ecoliner, Greenheart, Orcelle) [39]. Solar and wave energy will provide the energy source to produce green fuels such as hydrogen and ammonia.

### 4.1. Wind

Wind propulsion can be categorized into the following types [26,39]:

- Soft-sail,
- Fixed-sail,

- Flettner rotor,
- Kite-sail, and
- Turbine technologies.

The major drawback of wind is its intermittency and difficulty in maximizing the full power potential while sailing into the wind [39]. Wind can be used for primary and auxiliary propulsion. If wind is to be used for primary and auxiliary propulsion in existing ships, the structural stability has to be taken into account. It may even be the case that for existing ships, retrofitting some technologies is impossible, such as Flettner rotors.

Of course, wind can also be used (onshore or offshore) to electrolyze water in order to generate green hydrogen. There has been renewed interest in the production of green hydrogen by electrolysis of seawater by using electricity generated by an offshore wind turbine, an example is the joint press release of Siemens Gamesa and Siemens Energy announcing a "total investment of ~€120 million over five years in developments leading to a fully integrated offshore wind-to-hydrogen solution" [84].

### 4.1.1. Soft-Sails

Soft-sails attached on masts are a proven and mature technology, which can be deployed for either primary or auxiliary propulsion [39]. Soft-sails can be easily retrofitted into existing ships or included as part of a new ship design [39]. Examples are the Ecoliner, and Seagate delta wing sail, which is a patented folding delta wing sail [39]).

### 4.1.2. Fixed-Sails

Fixed sails are rigid sails attached on a rotating mast. Different fixed-sail designs have been used in ships such as the UT wind challenger and Ecofoil.

### 4.1.3. Flettner Rotors

Flettner rotors take advantage of the Magnus effect (propulsion when wind flows over a rotating cylinder). Flettner rotors were installed on E-Ship and Viking Grace. Deck space for different ship types is the best candidate for retrofitting Flettner rotors to bulkers and tankers up to VLCC class, which is being actively considered [39].

### 4.1.4. Kite Sails

Kite sails are attached on the bow of a ship and operate at altitudes so that they can utilize high wind speeds. The kite sail can help reduce annual fuel costs by 10–35% [85]. The MS Beluga Skysail, which was the first ever commercial container ship partially powered by a kite sail [39,86].

### 4.1.5. Wind Turbines

Apart from producing electricity (either onshore or offshore), wind turbines have not received much attention for ship propulsion. This is due to the fact that wind turbines suffer from inherent issues of vibration, stability and low energy efficiency conversion [39]. However, if wind turbines find their way for ship propulsion or for auxiliary power, their main advantage would be the harnessing of wind power even when the ship is sailing directly into the wind [39].

### 4.2. Solar

Photovoltaic (PV) cells produce electricity directly from the sunlight via the photo-electric effect. This is a fast-evolving technology with fast-paced progress, however, PV cells face two fundamental issues: (1) the lack of sufficient area on a ship to deploy the PV cells, and (2) the energy storage required, in the form of batteries [39]. Additional constraints of PV cells are the potential erosion of solar panels due to the presence of salt in the seawater [1] and intermittency issues. It is worthwhile mentioning that potential emissions reduction from solar energy is limited [8]. Smith et al. [11] assume a "reduction of around 0.1–3% of auxiliary engine fuel consumption". Bouman et al. [17] report the

potential in $CO_2$ reduction in the range of 0.2–12%. Even though there has been tremendous progress on batteries, ship propulsion via solar PV, on the other hand, requires further research and development. PV applications are likely to be confined to relatively small ships [81]. A solar ship that integrates PV cells into its power system is one of the most promising and fastest developing green ship systems [87].

Assuming an average improvement of the order of 1.5% on auxiliary fuel consumption and 6% on $CO_2$ potential reduction (based on the studies of Smith et al. [11] and Bouman et al. [17]) is quite significant and promising. Given the latest progress of PV systems in general (for example the continuous improvement in efficiency) and their rapid growth shows that they are the most promising technology utilizing renewable energy. However, PV systems on a ship require further R&D in terms of materials due to the corrosive nature of salty water, hence the potential of PV systems on ships is lower compared to on land applications.

Solar-Hybrid Systems

PV cells can be combined with fixed sails so that solar and wind energy are utilized at the same time [39]. Such an example is the OCIUS Technology's SolarSailor design [88]. Another example is Japan-based Eco Marine Power which is developing a large solar-sail Aquarius Marine Renewable Energy (MRE) for tankers and bulkers [89].

Batteries can also be used in conjunction with PV cells, an example is the Greenheart design that uses solar energy to "charge lead-acid batteries for auxiliary propulsion for its primary sail rig" [39]. However, solar (along with wave energy and wind turbines) can provide energy to produce green hydrogen by electrolysis of water. The hydrogen produced can then be later used in fuel cells [39] or ammonia production for later use in fuel cells.

*4.3. Biofuels*

Biofuels are produced from organic waste such as plant and animal waste [8]. At the moment, the main sources of biofuels are from plant-based sugars and oils, such as from palm, soybean and rapeseed [33]. Hsieh & Felby provide in a form of a flowchart an overview of the different feedstock conversion routes for marine biofuels, showing conventional and "advanced" biofuels ([33], p. 42).

Table 3 refers to first, second and third generation biofuels, the latter two known as "advanced biofuels", which depends on the source of carbon used [39]. The European Biofuels Technology Platform defines first, second and third generation biofuels as follows [39]:

- First Generation: "The source of carbon for the biofuel is sugar, lipid or starch directly extracted from a plant. The crop is actually or potentially considered to be in competition with food."
- Second Generation: "The biofuel carbon is derived from cellulose, hemicellulose, lignin or pectin. For example, this may include agricultural, forestry wastes or residues, or purpose-grown non-food feedstocks (e.g., Short Rotation Coppice, Energy Grasses)."
- Third Generation: "The biofuel carbon is derived from aquatic autotrophic organisms (e.g., algae). Light, carbon dioxide and nutrients are used to produce the feedstock, "extending" the carbon resource available for biofuel production."

Advanced biofuels are excellent candidates for the shipping sector and the most viable option as a renewable energy source [39]. Advanced biofuels "have very low sulfur levels and low $CO_2$ emissions, as such they are a technically viable solution to low-sulfur fuels meeting either the very low sulphus fuel oil (VLSFO) or ultra low sulphur fuel oil (ULSFO) requirements" [33]. Biofuels are currently the most relevant alternative for replacement or blending with gasoline (blended with bioethanol) or diesel (blended with biodiesel) in the transport sector [39]. However, usage and experience in the shipping industry is limited [39].

The challenge with biofuels in the shipping sector is that there is little experience and knowledge "on handling and applying biofuels as part of their fuel supply. Another challenge is that the volumes of biofuels required to supply the shipping sector are large" [33]. Hence, sustainable biofuel production is limited taking into account food price, natural resources (such as availability of land) and social conditions [8]. There are also concerns regarding the storage and oxidation stability of biofuels and further research is required [33].

Hsieh and Felby ([33], p. 74) present a SWOT analysis regarding marine biofuels. Compared to fossil fuels, biofuels have a higher cost, and this is expected to remain at least for the short- and medium-term. However, a combination of policies, regulations (reduced sulfur levels in marine fuels and reduction of GHG emissions), incentives and technology and infrastructure improvements may help create a healthy market for biofuels in the shipping industry [33].

### 4.3.1. Liquid Biofuels

Liquid biofuels can be combusted in a spark or compression ignition engine and are applicable to any vessel type [39]. A spark ignition engine can combust up to 10% bioethanol (E10) and a compression ignition engine can combust up to 20% biodiesel (B20) without significant adaptations. For blends of fuels between E10 and E85, the engine must be retrofitted with a fuel sensor in order to determine the proportion of ethanol in the blend and hence optimize the engine's performance. For proportions greater than E85, the engine performance is rather poor ([90], p. 507). On the other hand, compression ignition engines can combust blends in the range of B20 and B100 (i.e., 100% biodiesel) without too much retrofitting or degradation in engine performance. However, special care must be taken regarding the fuel due to its viscosity ([90], p. 507).

There have been tests on third generation biofuels in shipping. An example is the Maersk Kalmar, where Maersk partnered with the US navy to test algal biofuels [91]. The test included testing fuel blends (from 7% to 100%) in the auxiliary engine through a ~6500 nautical mile journey from Northern Europe to India. The test used Solazyme's 100% algal derived advanced biofuel, Soladiesel$_{RD}$, and proved that it is possible to substitute an advanced renewable fuel for diesel [92].

### 4.3.2. Biogas

Biogas can be either methane or hydrogen. Biogas is produced from anaerobic digestion within an enclosed environment which consists of microbes that break-down organic material ([90], p. 470). Biogas can be processed in order to remove impurities, hydrogen sulfide and moisture [39]. Methane produced from anaerobic digestion of organic waste can be liquefied to from Liquified Bio-Methane (LBM) [39]. The shipping industry favors LNG as a transitional fuel for decarbonization [39], however, LBM is seen as an option for deep decarbonization in the long-term because it is produced from renewable sources. Hydrogen can be either used in a fuel cell to generate electricity for propulsion or blended with the fuel for combustion in a gas turbine or internal combustion engine to achieve reduction in emissions.

The CE Delft [93] study investigated the availability and costs of LBM and Liquified Synthetic Methane (LSM), also known as e-methane. Note that in the CE Delft [93] study, it is assumed that LSM is produced from synthesis of $CO_2$ and $H_2$, where hydrogen is produced from renewable energy sources. The $CO_2$ is from "recycled $CO_2$", where it is either captured from industrial processes (e.g., flue gas) or the air [93]. The study determined that the future maximum conceivable sustainable supply of LBM and LSM exceeds the energy demand from the shipping sector, provided that biomass will be used to produce methane and sufficient investments are made in renewable electricity production [93]. In addition, the production costs of LSM and LBM may not be significantly higher and could even be comparable to the production costs of other low- and zero-carbon fuels [93]. If the costs of bunkering infrastructure and ships are comparable as well, then

LSM and LBM fuels may be viable candidates to achieve decarbonization in the shipping sector [93].

**Table 3.** Internal combustion engine types, manufacturer and engine model, as presented in Reference [94].

| Engine Type | Manufacturer | Engine Model |
|---|---|---|
| Four-stroke | Anglo Belgian Corporation (Gent, Belgium) | VDZC Series; DL36 Series |
| | Akasaka diesels (Tokyo, Japan) | AX 28 Series |
| | Caterpillar Inc (Deerfield, MA, USA) | 3500 Series; 3600 Series |
| | MaK Motoren GmbH and Co. (Peoria, IL, USA) | M601; M20C, M25C, M32C and M43C Series |
| | Daihatsu Motor Co. Ltd. (Ikeda, Japan) | DK-28 Series; DC-17Ae Series |
| | Deutz AG (Cologne, Germany) | TCD 2015 V Series |
| | Doosan Engine Co. Ltd. (Changwon, Korea) | V222TI; L136TI Series |
| | Hanshin Diesel (Kobe City, Japan) | LA 30; LA 34 Series |
| | Hyundai Heavy Industries Co. Ltd. (Ulsan, Korea) | HiMSEN Series H17/28; H21/32; H32/40 |
| | MAN Diesel and Turbo SE (Augsburg, Germany) | L23/30; L16/24; L21/31; L27/38; L32/40; L40/54; L48/60 CR; L58/64; L32/44 CR; D2876 |
| | Mitsui and Co. Ltd. (Tokyo, Japan) | ADD30V |
| | MTU Friedrichschafen GmbH (Friedrichshafen, Germany) | MTU V 4000 |
| | Paxman (Colchester, UK) | VP185 |
| | Rolls-Royce Group Plc (London, UK) | B32:40R; C25:33L |
| | Ruston and Hornsby (Lincoln, UK) | RK 270; RK 280 |
| | Pielstick (Augsburg, Germany) | S.E.M.T. Pielstick series PC2.6B |
| | Sulzer Brothers Ltd. (Winterthur, Switzerland) | Z40; ZA40S |
| | Wärtsilä NSD (Helsinki, Finland) | L20; L/V32C; L/V46C; 64C |
| | Yanmar Co. Ltd. (Osaka, Japan) | 6N21; EY18L; EY26L; AYM |
| Gas four-stroke | MAN Diesel and Turbo SE (Augsburg, Germany) | V35/44G |
| | Mitsubishi Heavy Industries (Tokyo, Japan) | KU30G |
| | Rolls-Royce Bergen (Hordvik, Norway) | C26:33LPG |
| Gas-diesel dual-fuel four-stroke | MAN Diesel and Turbo SE (Augsburg, Germany) | 35/44DF |
| | Caterpillar Inc (Deerfield, MA, USA) | M46DF |
| | Wärtsilä NSD (Helsinki, Finland) | L50DF |
| Two-stroke ship low-speed crosshead | MAN Diesel and Turbo SE (Augsburg, Germany) | MC, MC-C, ME, ME-C and ME-B series |
| | Mitsubishi Heavy Industries (Tokyo, Japan) | UEC LSH-Eso; LSE-Eco |
| | Wärtsilä-Sulzer (Helsinki, Finland) | RTA; RT-Flex Series |
| | WinGD (Winterthur, Switzerland) | W-X Series |
| Gas diesel two-stroke ship low-speed (low pressure) | WinGD (Winterthur, Switzerland) | X-DF Series |
| Gas diesel two-stroke ship low-speed (high pressure) | MAN Diesel and Turbo SE (Augsburg, Germany) | ME-GI Series |

## 5. Maturity of Technologies

This section discusses various technologies used to provide power for propulsion as well as auxiliary and ancillary applications.

### 5.1. Internal Combustion Engines (ICE)

Internal combustion engines (ICE) provide power to a ship (for auxiliary and ancillary applications) and power for propulsion. These technologies can be coupled with the technologies discussed in Section 3 to reduce fuel consumption as well as the alternative fuels discussed in Section 2 to reduce/eliminate emissions (either GHG or pollutant emissions).

Marine ICEs have a leading position in the global merchant fleet: almost 98% of the merchant fleet toady employ marine ICEs [94]. Currently, ICE engines in shipping have the highest efficiencies [94]. In addition, the reduction of cost of energy because of the use of heavy fuel oil [94] has made the ICE very attractive for shipping. Moreover, the transition of the gas and gas-diesel cycles and the introduction of new technologies related to control systems have considerably improved their environmental performance [94].

In terms of maturity, ICE engines have been around for more than a century and hence, knowledge and expertise on this field is vast. Thus, the operation of ICE is simple, advanced and robust, with a long lifetime, sometimes equivalent to a ship's lifetime [94].

By improving ICEs, the shipping sector will be able to reach its emissions targets. This is because of ICEs flexibility to accommodate future alternative fuels, its dominance within shipping and its ability to be further improved and tuned with smart control methods [95].

5.1.1. Types of Engines Used in Shipping

Currently, there are numerous types of ICE and engines as well as multiple ICE manufacturers, as summarized in Table 3. The main types of marine engines are four-stroke engines, gas and gas-diesel four-stroke, two-stroke ship low-speed crosshead engines and gas-diesel two-stroke low-speed engines.

Four-Stroke Engines

Four-stroke engines are the most common engines in the marine sector that are used on vessels of various sizes ([94], pp. 1–2). Four-stroke engines compared to two-stroke engines are smaller in size and weight as well as cost of construction. On the other hand, the cylinder heads of four-stroke engines have a more complicated design due to the presence of camshafts, valves, etc. ([94], pp. 1–2). The development of high-pressure fuel injection, optimized mixing air and fuel in the combustion chamber, and new approaches in the management of medium- and high-speed engines, has made four-stroke engines environmentally friendlier, especially in $NO_x$ emissions ([94], pp. 1–2). Therefore, four-stroke engines are attractive for technical fleet vessels, passenger and cargo ferries and cruise ships, operating in areas with current restrictions on emissions of harmful substances ([94], pp. 1–2), such as emission control area (ECA) zones.

Two-Stroke Engines

The advantage of two-stroke engines compared to four-stroke engines is their ability to increase the power output for the same working volume ([94], p. 187). To achieve a high stroke to bore ratio, low speeds are required to limit the maximum piston speed, hence the need for a crosshead design and uniflow-scavenging concept [96]. The prime mover of merchant ships has been, for more than a century, the marine diesel engine, which is nowadays predominantly a low-speed, two-stroke, crosshead-type, reversible, uniflow-scavenged, turbocharged, electronic engine. Due to the size and fewer moving parts, two-stroke engines are the most thermally efficient and most reliable engines. In addition, marine diesel engines are turbocharged because they are force-scavenged [96].

The presence of a large amount of gas-air mixture increases the risk of an explosion, which can be catastrophic onboard a ship. Therefore, in low-speed two-stroke engines, mixing of air and fuel is achieved via two approaches ([94], p. 374):

- Fuel (in the gas phase) enters the combustion cylinder after the exhaust valve is closed at the initial stage of the compression stroke under relatively low pressure; hence, such systems are called low-pressure supply systems.
- Fuel (in the gas phase) with the ignition fuel enters the combustion cylinder at the end of the compression stroke which is at high pressure; hence, such systems are called high-pressure supply systems or direct gas injection (GD).

Gas and Gas-Diesel (Dual Fuel) Engines

To achieve low emissions levels, especially in ECA zones, the use of gas as fuel is the most promising way. Gas fuels reduce emissions in comparison to heavy fuel oil: $SO_x$ emissions are non-existent (due to the absence of sulfur in the fuel), $NO_x$ emissions are reduced by 90% and PM and $CO_2$ emissions are reduced by 30%. The most promising gas fuels are methane, propane and butane. Onboard a ship, methane is stored in liquefied state in cryogenic tanks as LNG, whereas propane-butane mixtures are stored in liquefied state as LPG at ambient temperatures and high pressures in tanks ([94], p. 167).

There are three different approaches where gas fuels are used in either two- or four-stroke engines ([94], p. 168):

(1) Convert diesel engines to operate according to the Otto cycle, i.e., mix the air-fuel externally to the combustion cylinder and ignite the air-fuel mixture with electric spark ignition.

(2) Mix the air-fuel externally to the combustion chamber and ignite the air-fuel mixture with electric spark ignition combined with liquid fuel injection into the combustion cylinder: DF engine.

(3) Mix the air-fuel in the combustion chamber and ignite the air-fuel mixture with electric spark ignition combined with liquid fuel injection into the combustion: DF engine.

The first two approaches are used in four-stroke engines for various applications, whereas the latter approach is used in low-speed, two-stroke engines ([94], p. 168). The first approach is more often used in stationary power applications, such as provision of power to offshore platforms, but is less common in ships ([94], p. 169). The second approach is used in medium- and high-speed engines ([94], p. 177).

Currently, most engines work on a dual-fuel (DF) basis, either two- or four-stroke, in order to accommodate for variations in the ship's operating conditions, provide independence of cargo type and size and maintain the engine's ability to operate solely on liquid fuels or a blend of gas-liquid fuel. Hence, DF four-stroke engines have the ability to operate on gas fuel, liquid fuel or a combination, and in different proportions ([94], p. 167). The performance and emissions of DF engines vary depending on operating conditions and the level of sophistication of the control system. Generally, DF engines perform best under moderate to high load conditions and can often have equal or better fuel-efficiency as a pure diesel under the same conditions [97].

DF engines have proven to be a viable candidate for achieving environmental performance, future energy transformation and the reliability demanded of marine engines. There are, however, unsolved problems and technical challenges so that DF engines may become a key engine for marine applications [98]. These challenges, as mentioned by Ohashi [98], are:

• Effective lubrication (and the varying degree of sulfur level) because of the use of different fuels. The knocking margin due to alkali and other compounds depositing on the cylinder walls, hence it will be difficult to select a suitable lubricant that satisfies operation in diesel or gas mode.

• Different characteristics of natural gas at different bunkering stations (varying degree of humidity, $CO_2$, etc.) which may cause knocking.

### 5.1.2. Combustion of Alternative Fuels in ICE

This section discusses the use of alternative fuels in ICE, namely LNG, hydrogen, methanol and ammonia.

### LNG in ICE

Diesel-LNG engines have many advantages, because LNG has a better C:H ratio, and thus reduced $CO_2$ emissions. In addition, LNG originates from methane, which allows diversification of fossil fuel supplies. Under standard temperature and pressure conditions, LNG is a gas, hence LNG vaporizes, mixes and burns much better than diesel, thus achieving reduced emissions of regulated pollutants [99].

LNG is burnt in dual-fuel engines, either in low-pressure engines or in high-pressure engines [74]. In low-pressure engines, the occurrence of methane slip is higher compared to high-pressure engines [74]. However, $NO_x$ in low-pressure engines is low compared to high-pressure engines, hence the need to use selective catalytic reduction (SCR) for the latter case [74]. Depending on engine conditions, CO and unburnt HC may increase [76]. Li et al. [76] in their study retrofitted an existing marine diesel engine with control systems in order to test the engine's performance with LNG and concluded that "dual fuel control system could work steadily over a long period of time" [76]. Hence, a challenge in retrofitting engines with LNG is the successful implementation (and testing) of a dual-fuel

control system with intelligent switching mechanisms for diesel fuel mode and dual-fuel mode and that will ensure steady and reliable operation of the engine. An additional challenge of LNG is the potential knock when used in dual-fuel engines due to its higher flammability [100].

Hydrogen in ICE

Green hydrogen is an excellent candidate for the deep decarbonization of the shipping industry. However, hydrogen has several challenges for its use as a fuel in ICE. For example, hydrogen has a very low minimum ignition temperature (see Table 4) which may cause uncontrollable pre-ignition events and high combustion temperatures by burning hydrogen-air mixtures close to the stoichiometric composition, leading to high $NO_x$ emissions [101]. However, hydrogen has a high autoignition temperature and hence it is more suitable as a fuel for a spark-ignition engine rather than a compression-ignition engine [101], which can be an issue for the shipping sector (because ICEs are predominantly compression-ignition engines). However, the use of hydrogen as secondary fuel in compression-ignition engines is preferred because lower emissions with little penalty in engine performance is incurred [102].

**Table 4.** Fuel properties of ammonia, hydrogen and gasoline. Reproduced from Reference [103], data were obtained from various sources [104–106].

|  | Unit | Ammonia | Hydrogen |
|---|---|---|---|
| Lower heating value | MJ/kg | 18.8 | 120.0 |
| Flammability limits, gas in air | Vol. % | 15–28 | 4.7–75 |
| Laminar flame speed | m/s | 0.015 | 3.51 |
| Autoignition temperature | °C | 651 | 571 |
| Absolute min. ignition energy | mJ | 8.0 | 0.018 |
| Octane rating, RON | - | >130 | >100 |
| Density, 25 °C, 1 atm | g/L | 0.703 | 0.082 |

There are examples of ships that combust hydrogen such as the Hydroville by CMB, a 16-passenger shuttle. The technology is a hybrid engine that allows it to run on diesel and hydrogen [107]. In addition, the CMB technology is building several hydrogen-powered vessels. "Economics and environmental benefits of low-carbon solutions in oceanic transport provide the opportunity to compete in the maritime market with the potential to virtually eliminate pollution at the point of use" [107].

Ammonia in ICE

Ammonia is an attractive fuel because of the absence of carbon and sulfur atoms in its chemical formula. As shown in Table 4, the octane rating of ammonia and hydrogen is higher compared to gasoline, making it preferable to run at a higher compression ratio [103] and hence ideal for diesel engines. Ammonia has a high autoignition temperature; hence, to overcome this issue, the dual-fuel approach in a diesel turbocharged multicylinder engine [108] may be the way forward.

Note that the energy content of ammonia is lower than that of a hydrocarbon fuel on a mass basis. However, the stoichiometric air–fuel ratio for ammonia is significantly lower than that of a typical hydrocarbon fuel (6 compared to 14 for diesel and 15 for gasoline). Hence, more ammonia is able to combust with the same amount of air [108].

The market has recently shown signs of interest for the use of ammonia as a fuel in ICE (both the shipping sector and the automotive sector). For example, Wärtsilä with Knutsen OAS Shipping AS, Repsol and Sustainable Energy Catapult Centre are collaborating and are planning to test ammonia in a marine four-stroke engine, under the Norwegian Research Council through the DEMO 2000 program [109]. Wärtsilä has already performed some initial tests of ammonia in dual-fuel and spark-ignited gas engines, which "will be followed by field tests in collaboration with ship owners from 2022, and potentially also with energy

customers in the future" [110]. There are additional initiatives from MAN Energy solutions who "claim that their dual-fuel engine developed for LPG may use liquid ammonia in a dual-fuel setup" [57,111]. In addition, "MAN Energy Solutions, Shanghai Merchant Ship Design and Research Institute (SDARI) and American Bureau of Shipping (ABS) have a development project for an ammonia-fueled feeder container vessel intended to use this technology" [57].

Ammonia has a number of properties that require further investigation before its commercial exploitation [110]. These are:

- Poor ignition and very slow flame propagation speed compared to other fuels, see Table 4 for minimum ignition energy and laminar flame speed.
- Toxic and corrosive, thus the requirement of sustainable safety and storage solutions.
- Higher $NO_x$ emissions, unless controlled either by after-treatment or by optimizing the combustion process.
- Regulations/policies will need to be developed for its use as a marine fuel.

Tests from industry in the next few years will determine whether ammonia will hold it's promise as the next generation of fuel (either as a single or as a dual fuel) for ICE applications given the aforementioned constraints, unknowns and issues.

Methanol in ICE

Methanol as fuel can be used both in marine spark-ignition (SI) and compression-ignition (CI) engines, with the latter in dual-fuel mode [63]. Methanol can be directly compliant with IMO 2020 global sulfur cap without additional measures in marine engines and can lead to zero PM emissions. There is a significant reduction in $NO_x$ emissions, provided that exhaust gas recirculation (EGR) or water emulsion is used in dual-fuel engines [63].

Methanol use in existing vessels has considerably lower retrofit costs compared to LNG retrofit costs. Specifically, the methanol retrofit costs range from 25% to 35% of the corresponding LNG retrofit costs for 10–25 MW engines [63].

Andersson & Salazar [65] show the necessary retrofitting (and amount of pipework required) to introduce methanol into the combustion cylinder. The retrofits where part of the Stena Germanica which used a medium-speed four-stroke engine [65]. A two-stroke engine has also been developed and tested for new-build tankers by MAN Diesel and Turbo [65,112]. The cost of retrofitting the engine of a ship from "diesel fuel to dual-fuel methanol/diesel fuel has been estimated to be €250–350/kW for large engines (10–25 MW)". The corresponding cost in retrofitting LNG is of the order of €1000/kW. However, the actual cost for installing fuel tanks and supply will depend on the layout of an individual ship [65].

5.1.3. After-Treatment Technologies in ICE

After-treatment technologies cannot be ignored from the discussion of decarbonization of the shipping industry because we are interested in green shipping. Marine diesel engines will be used in the shipping industry in the foreseeable future and hence the discussion should include after-treatment technologies. Often, emissions (pollutant and GHG emissions) are interlinked with each other and most likely competing (for example, an improvement in $NO_x$ emissions will have opposite effects on CO emissions). A major issue with diesel engines in general is their high $NO_x$, soot and particulate matter emissions. This section explores the various after-treatment technologies available for diesel ICEs.

There are different after-treatment strategies (and combinations) that can be used for reducing emissions from ICEs. These strategies, together with recent advances in after-treatment technologies, involve reduction with and without filter, reduction with catalyst and without catalyst and exotic after-treatment techniques such as plasma-assisted techniques, $NO_x$ and soot combined reduction [113]. Strategies to reduce emissions and pollutants come at cost and/or fuel consumption penalty [114]. The flowchart in Figure 2 illustrates the various methods available for after-treatment regarding $NO_x$ reduction.

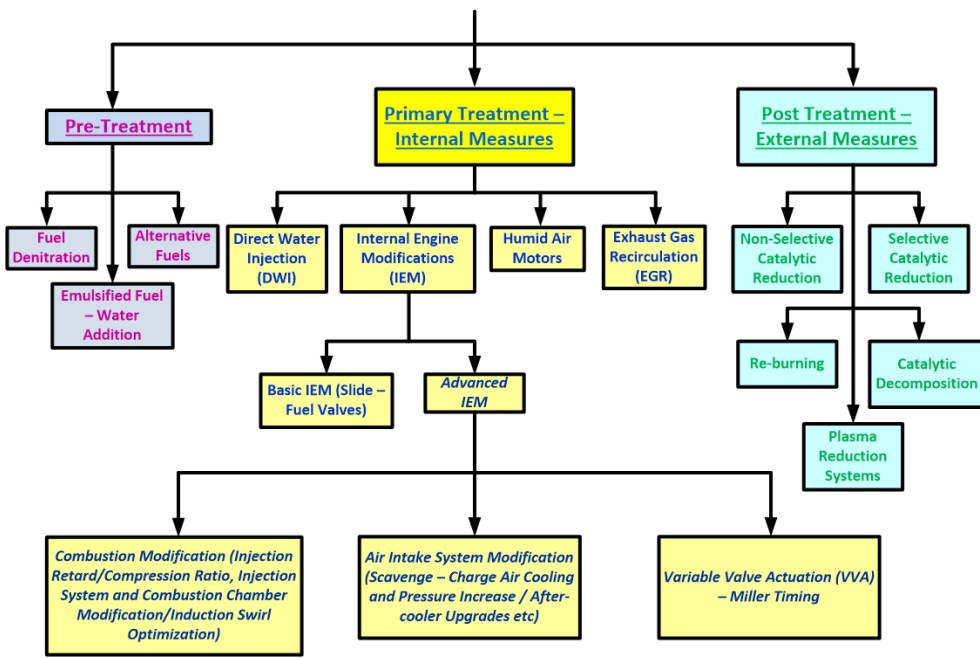

**Figure 2.** Marine diesel $NO_x$ reduction methods (image obtained from Reference [115]).

Direct Water Injection (DWI)

With direct water injection (DWI), water is directly injected in the combustion cylinder, thus achieving high $NO_x$ reduction (~50%) [116] since peak temperatures are reduced. Compared to other Wetpac methods (humidification and water-in-fuel emulsions), DWI achieves a low water consumption and water quality is a less critical issue. However, this method has higher fuel consumption with increased smoke, especially at low loads [116]. DWI can improve engine performance and hence reduce $CO_2$ emissions. Zhang et al. [117] in their study mention an improvement of ~4% in thermal efficiency.

Intake Air Humidification

As opposed to DWI, the intake air can be humidified, before it enters the combustion cylinder. Compared to DWI, this method achieves lower $NO_x$ reductions (10–40%) with a higher water consumption. However, this system compared to DWI is less complicated and cheaper [116]. In addition, this system can be used to achieve a higher knock margin for gas engines [116].

Water-in-Fuel Emulsion

Fuel and water droplets are homogenized before injection in the combustion cylinder. This method has a low $NO_x$ reduction potential (10–25%) and limited flexibility due to increased smoke formation and reduced engine performance. In addition, if standard fuel injectors are used, they exhibit increased mechanical stress [116].

Non-Thermal Plasma (NTP)

In the NTP after-treatment technique, the exhaust gas is ionized (also known as plasma) with the aid of electricity. The plasma contains highly energetic electrons that convert oxygen into oxygen radicals. The oxygen radicals then react with NO and are converted into $NO_2$. Then, $NO_2$, with the aid of a catalyst, is converted into $N_2$ [113,118,119]. Note that plasma can be generated either by an electron beam or by electric discharge [113].

Selective Catalytic Reduction (SCR)

In an SCR, a catalyst is used to reduce the activation energy required to perform $NO_x$ reduction. SCR systems generally have the best $NO_x$ removal efficiency compared to

other de-$NO_x$ methods [113]. A reductant (usually ammonia by injecting aqueous urea) is introduced in the exhaust gases, whereby $NO_x$, with the help of a catalyst, is reduced to $N_2$. The SCR technology, according to the review study of Lu et al. [120], is the most promising technology for meeting the IMO $NO_x$ Tier III limits. There are various SCR treatment philosophies in terms of reductant used [113], namely:

- Ammonia SCR after-treatment
- Hydrocarbon SCR after-treatment
- Carbon monoxide SCR after-treatment
- Hydrogen SCR after-treatment
- Alcohol SCR after-treatment

$NO_x$ Trap After-Treatment

A $NO_x$ after-treatment device uses a base metal oxide and a noble metal for reduction of $NO_x$. The function of the base metal is to trap $NO_2$ during lean operation and release it during rich operation. The function of the noble metal is to aid the oxidation and reduction chemical reactions [113].

Exhaust Gas Recirculation (EGR)

$NO_x$ reduction can be achieved by exhaust gas recirculation (EGR) by injecting exhaust gases into the combustion cylinder. Exhaust gases, due to their high specific heat capacity, lower the peak temperatures, thereby reducing $NO_x$. Temperature reduction also occurs due to the lower $O_2$ content in the engine [120]. EGR can achieve up to 50% $NO_x$ reduction compared to a conventional diesel engine [121]. EGR combined with scrubbers can achieve $SO_x$ and PM emissions reductions and, as an additional benefit, protect the EGR cooler from clogging [121]. However, EGR may increase the installation complexities and their cost with higher fuel consumption [121]. In addition, EGR generates wastewater, which needs to be taken into account when designing such a system [120]. A typical EGR system (shown in Reference [120]) where the exhaust gases are purified, cooled and dehumidified by bypassing part of the exhaust gas to the washing tower. Then, the treated exhaust gas is pumped through the blower mixed with fresh air and then enters the engine [120].

Control of Particulate Matter (PM) Emissions

The role of the Diesel Oxidation Catalyst (DOC) oxidizes CO and unburned hydrocarbons to $CO_2$ [113]. Effectively, a DOC is a stainless-steel tube, which within it contains a honeycomb monolith structure. When exhaust gases, with the aid of a platinum catalyst, pass through the DOC, carbon monoxide, unburned hydrocarbons and nitrogen oxides are oxidized [113]. Diesel engines also contain a Diesel Particulate Filter (DPF), a ceramic monolith wall filter. When exhaust gases pass through the DPF, soot particles stick on the filter [113]. However, the DPF filter may become clogged due to its continuous use, hence the need to oxidize soot [113]. To achieve oxidation of the soot particles, higher temperatures are required (~550 °C, whereas the exhaust gas temperatures are of the order of 200 °C) [113], hence the need of active or passive (also known as continuous regeneration traps (CRT)) regeneration methods.

De-$SO_x$ After-Treatment Technologies

Exhaust gas treatment systems (EGTS) are the most promising technologies to offer solutions which will comply with the IMO's $SO_x$ and PM emission regulations [122]. The main EGTS technologies currently used today are wet scrubbers (either open-loop with seawater or closed-loop with aqueous NaOH solution) [123]. In addition, there are hybrid aqueous scrubber systems that use a combination of open- and closed-loop systems. Finally, there are dry flue gas scrubbers. All these technologies can be used to achieve $SO_x$ and PM reductions. Scrubbers are the most economical and can meet the "IMO $SO_x$ Compliance Independent of Fuel Sulphur Cap", and a mature technology ready to be installed on new or existing vessels, but they are complex, heavy and large installations [124].

*5.2. Fuel Cells*

Fuel cells are an efficient way of producing low-carbon electricity and they are a key technology to unlock the use of future alternative fuels [1,125]. This is because fuel cells may provide a suitable solution, due to the fact that they are fuel-efficient and emit few hazardous compounds [126]. Fuel cells convert fuel into electricity, and it is a proven technology for land-based power that could replace internal combustion engines [4,127]. Staffell et al. [125] mention that the cost of fuel cells could converge with the cost of ICEs and electric vehicles (EVs) by 2030 [128].

Although electricity is used for auxiliary uses in ships, recent trends have shown that electricity can be used for propulsion [126]. Fuel cells can improve energy conversion efficiency to over 60%, and if waste heat is used, an 80% efficiency can be achieved [54,127]. The improvement in efficiency will reduce the need of an auxiliary power, which may lead to improved ship design with additional space for cargo [54], bearing in mind that this is a scenario without heat recovery (which can cause major issues with a fuel cell's power concentration). However, currently, fuel cells are more expensive compared to internal combustion engines (ICE) on a $/kW basis [54]. On the other hand, Ballard demonstrates a cost reduction of the order of 70–80% for land-based vehicles utilizing fuel cells [129], which could be applied for the shipping industry. It is worthwhile mentioning that once fuel cells are installed, their operation and maintenance costs are low because there are very few moving parts compared to ICEs [54,130].

In 2008, the Zemships (Zero Emissions Ships) project developed the FCS Alsterwasser, the first fully PEMFC fuel cell-powered ship, which was powered by two hydrogen fuel cell units of 48 kW power capacity [39,131]. It was a 100-passenger vessel for inland waterways, and a number of other small ferries and river boats have followed suit. The ship operated until late 2013, but the challenge of economically operating the hydrogen charging infrastructure made it unfeasible [39]. In 2013, the FellowSHIP project, which ended in 2018, successfully tested a 330 kW MCFC fuel cell on the Viking Lady as an auxiliary power unit [39,131]. This was the first fuel cell test on a merchant ship with an electrical efficiency of ~44.5%. When heat recovery was retrofitted, the corresponding efficiency increased to ~55%, with further improvements possible [39]. In 2012, Germanischer Lloyd tried out design concepts for zero-emissions using liquid hydrogen as fuel to generate power with a combined fuel cell and battery system. However, the sustainability of hydrogen production is a critical issue [39]. For the interested reader, van Biert et al. [126] provide an overview of noticeable maritime fuel cell application research projects and lessons learned up to 2016, and Xing et al. [131] provide a list of successful fuel cell projects in maritime applications since 2000, as summarized in Table 5.

**Table 5.** List of successful projects as noted by Xing et al. [131]. * Used fuel cells as auxiliary power units (APU) or as supplement to main propulsion power.

| Fuel Cell Type | Project/Vessel Name | Fuel | Capacity |
|---|---|---|---|
| AFC | Hydra | Metal hybride | 6.9 kW |
| | Hydrocell Oy | Metal hybride | 30 kW |
| LT-PEMFC | Elding | $H_2$ | 10 kW |
| | ZemShip Alsterwasser | $H_2$ | 96 kW |
| | Nemo $H_2$ | $H_2$ | 60 kW |
| | Hornblower Hybrid | $H_2$ | 32 kW |
| | Hydrogenesis | $H_2$ | 12 kW |
| | SF-BREEZE | $H_2$ | 120 kW |
| | Cobalt 233 Zet | $H_2$ | 50 kW |
| | US SSFC | Diesel * | 500 kW |
| HT-PEMFC | Pa-X-ell MS Mariella | Methanol | $2 \times 30$ kW |
| | RiverCell | Methanol | 250 kW |
| | MF Vågen | $H_2$ | 12 kW |
| | RiverCell ELEKTRA | $H_2$ | $3 \times 100$ kW |
| MCFC | MC WAP | Diesel * | 150/500 kW |
| | FelloSHIP Viking Lady | LNG * | 320 kW |
| | US SSFC | Diesel * | 625 kW |
| SOFC | METHAPU Undine | Methanol | 20 kW |
| | SchIBZMS Forester | Diesel * | 100 kW |
| | FELICITAS subproject 2 | LNG * | 250 kW |

## 5.2.1. Fuel Cell Types

There is a large variety of fuel cells, but in this report, the following fuel cells will be considered:

- Alkaline fuel cell (AFC),
- Low- and high-temperature polymer electrolyte membrane fuel cell (LT/HT-PEMFC),
- Phosphoric acid fuel cell (PAFC),
- Molten Carbonate fuel cell (MCFC),
- Direct Methanol fuel cell (DMFC) and
- Solid oxide fuel cell (SOFC).

Table 6 summarizes the aforementioned fuel cells with their corresponding temperature range, fuel requirements and possibility for internal reforming. Xing et al. [131] in their review argue that PEMFC, MCFC and SOFC are the most promising options for maritime applications, "once energy efficiency, power capacity and sensitivity to fuel impurities are considered". They also mention that currently, the main issues with applying fuel cells to maritime applications are (1) power capacity, (2) costs and (3) lifetime of the fuel cell stack [131]. Xing et al. [131] conclude that the way forward for fuel cell applications in the maritime industry (i.e., to improve their performance) is to couple fuel cells with batteries, modularization and optimized control and operating strategies. Note that in some fuel cell types (such as MCFC), waste heat recovery systems are applied to improve the overall efficiency [131,132], possibly coupled with organic Rankine cycles due to the low-grade temperatures.

**Table 6.** Commonly applied fuel cells, their temperature range, fuel requirements and the opportunity to reform fuel directly in the fuel cell, power capacity and main drawbacks. Data obtained from References [126,131].

| Fuel Cell Type | Operating Temperature (°C) | Fuel | Internal Reforming | Power Capacity | Drawbacks |
|---|---|---|---|---|---|
| AFC | 60–200 | $H_2$ | No | ≤500 kW | $CO_2$ poisoning |
| LT-PEMFC | 65–85 | $H_2$ | No | ≤120 kW | CO + S poisoning |
| HT-PEMFC | 140–220 | $H_2$ | No | ≤500 kW | CO + S poisoning |
| PAFC | 140–200 | $H_2$, LNG and methanol | No | 120–400 kW | CO + S poisoning |
| DMFC | 75–120 | methanol | No | ≤5 kW | Methanol crossover |
| MCFC | 650–700 | $H_2$, CO | Yes | 120 kW–10 MW | S poisoning, cycling effects, long start-up time |
| SOFC | 500–1000 | $H_2$, CO | Yes | ≤10 MW | S poisoning, cycling effects, mechanically fragile, long start-up time |

Out of all the fuel cell technologies, LT-PEMFC has seen the most rapid development, and hence has achieved high power densities and good transient performance. However, LT-PEMFC operates at low temperature and therefore requires a platinum catalyst in order to catalyze the electrochemical reaction [133]. PEMFC membranes require a wet membrane to facilitate the transfer of protons, which complicates water management due to the pores of the gas diffusion layer, especially for LT-PEMFCs [134]. At low temperatures, LT-PEMFC have low tolerance on impurities, such as CO, that deactivates the catalyst due to its high surface adsorption by the catalyst [135,136].

The membrane of the PAFC is a silicon carbide matrix saturated with phosphoric acid [126]. The higher operating temperatures offer a two-fold advantage: (1) reduces the need for a high platinum load, and (2) increases the CO tolerance. However, PAFC have low power density and low robustness [126]. As a response to these challenges, the HT-PEMFC combines a polymer electrolyte and a phosphoric acid membrane [137,138].

As shown in Table 6, for high-temperature fuel cells, CO becomes a fuel rather than a contaminant and expensive platinum catalysts can be replaced with cheaper nickel catalysts. Another advantage is the opportunity to use high-temperature waste heat and steam. MCFC are relatively mature high-temperature fuel cells and are commercially available, but still have issues with high cost, limited lifetime and low power density [139,140]. Low-temperature SOFCs have electrical efficiencies ~60% and are used as stand-alone units, whereas high-temperature SOFCs have efficiencies over 70% and are used in combination with gas turbines [141–143]. SOFCs have mechanical vulnerabilities, high costs and limited adoption [144].

### 5.2.2. Hydrogen as Fuel for Fuel Cells

The advantages of using hydrogen as a fuel, especially green hydrogen, have already been discussed in earlier sections of this review paper. The use of hydrogen in fuel cells has attracted lots of commercial interest. Recently, a Danish–Norwegian project has aimed to build and test a hydrogen-fueled ferry, the Europa Seaways operated by DFDS ferries, which has recently gained EU funding. The project aims to have Europa Seaways operational by 2027. In this project, several key players in the shipping and energy sectors have joined forces to build a ferry that will be able to transport 1800 passengers [145].

### 5.2.3. Ammonia as Fuel for Fuel Cells

Interest in using ammonia as a fuel for fuel cells in maritime applications is growing [146]. ShipFC is a funded project by Fuel Cells and Hydrogen Joint Undertaking (FCH JU) under the EU's Horizon 2020 research and innovation program, which was awarded

€10 million to install maritime fuel cells running on green ammonia. The project will use Viking Energy, an offshore vessel owned and operated by Eidesvik, as a test pilot [147].

### 5.3. Electric/Hybrid Propulsion

Energy storage for supplying power to zero-carbon electric propulsion can come from [8]:

- Batteries,
- Flywheels, and
- Supercapacitors.

Propulsion can come from electric motors, which are relatively cheap, however, combined with the cost of batteries and their housing on ships, it can be an expensive option [8]. Lloyd's Register and UMAS [148] compared, under different scenarios, the electric vessel with alternative fuels such as hydrogen, ammonia and biofuels, and estimated that electric vessels are the least profitable technology.

Hybrid marine engines are attractive because they can be fueled by diesel, LNG or hydrogen, and use a fuel cell, batteries or an electric motor [149]. Hybridization can offer 10–40% fuel savings and "payback times as low as one year" [39,150]. Hybrid propulsion also allows design flexibility in order to satisfy financial and environmental considerations of the operator [81].

### 5.4. Batteries and Supercapacitors

The costs of battery technology for electric vehicles (BEV) are rapidly falling, which may suggest that this type of technology might "become a more viable and readily available option also for other transport sectors such as shipping" [8]. Bloomberg New Energy Finance [151] estimates a price decrease for every doubling of capacity of ~19% between 2010 and 2016 for electric vehicles and stationary storage. The annual prices dropped even further than predicted in 2017–2020 and reached an average price of $137/kWh [152]. In fact, there are reports that the prices have fallen even lower than $100/kWh [152]. It is estimated that by 2030, lithium-ion battery pack prices will be $73/kWh [8]. On the other hand, Lloyd's Register and UMAS [36,153] in their various transition pathways investigation mention that "batteries play a minor role as a primary energy store/source onboard ships" for deep-sea shipping due to high costs and relatively low energy volumetric density. In fact, Lloyd's Register and UMAS [153] mention that in most of the cases they investigated, "the cost of batteries (cost of storage system) appears to be prohibitive relative to other zero-carbon options".

Zerocat 120 is the world's first lithium battery-powered ship, which is a 120-car ferry and a capacity of up to 360 passengers for short routes (~20 min), with a very short battery charging time (just ten minutes) [39,154].

Supercapacitors can also provide electric power to ships, but when compared to batteries, can store and release large amounts of electricity very quickly [8]. As part of the Ecocrizon research and development program, Ar Vegan Tredan [155], a zero-emissions passenger ferry powered with supercapacitors, was developed. The supercapacitors can be charged at portside (four minutes charging time), however, "this can only be considered a renewable energy-powered vessel" only if electricity originates from renewable sources (such as wind, solar, etc.) [39].

### 5.5. Gas Turbines

Gas turbines have found use in military vessels due to their potential to develop high speeds and short start-up times. Usually, engines from airplanes are retrofitted onto ships by modifying the inlets and outlets of the system. Gas turbines are expensive with high O&M costs. From an environmental standpoint, gas turbines have low efficiency.

Gas turbines have been tried on cruise ships, were the main factor for their choice was less noise and vibration, hence more comfort for the passengers. In addition, gas turbines have a high power to size ratio, hence the extra space can be used for additional passenger

cabins. Furthermore, gas turbines can operate with higher speeds, thus a cruise ship can reach its destination faster. The extra cost incurred was offset with more passengers.

### 5.6. Nuclear

Nuclear propulsion of ships has been around since 1955, mostly for military and submarine applications [8]. Nuclear propulsion is attractive because of its high-power density and low, stable prices, and low emission of GHG. Another important advantage of nuclear propulsion is the ability to operate over long periods without the need of refueling, which increases autonomy and provides independence to fuel price fluctuations [1,8,156]. Nuclear propulsion has been proven to be important for routes via the Russian Arctic, especially for icebreakers, both in terms of technicality and economics [156], whereby high-power demand is needed [157].

Nuclear propulsion is achieved with an onboard nuclear plant heating steam that drives steam turbines and generators. As already mentioned, nuclear propulsion is unlikely due to "political reasons", because, for example, port authorities will be unwilling to accept foreign ships with a nuclear reactor onboard [53]. Other issues against nuclear propulsion are legislation, training and safety against accidents, terrorism and non-proliferation [1]. For instance, the nuclear propulsion will need to adhere to the IMO's Resolution A.491-XII [158]: "which defines specific safety issues and criteria concerned with the protection of people and the environment from possible radiation hazards throughout the vessel's lifecycle" [26,158]. An additional challenge with nuclear propulsion is radioactivity of uranium fuel and waste which poses serious environmental and health hazards. This means "a complete overhaul" of the design of merchant ships because it will be driven by safety rather than efficiency [8]. With current focus in reducing GHG emissions, nuclear in the future may refocus in providing ships with hydrogen [156,159].

Small Nuclear Reactors (SMR) are an attractive option for shipping applications because they are compact, modular, safe and proliferation-free [26]. "The SMR technology is under development, with a thermal power output of over 68 MW and can be treated as a plug-in 'nuclear' battery. The use of an SMR for ship propulsion is an exciting prospect" [26].

### 5.7. Carbon Capture and Storage (CCS)

Carbon capture and storage is a very promising way to reduce $CO_2$ emissions. There are innovative ideas and active research to utilize captured $CO_2$ into a fuel, such as methane. For example, there is active research to develop a novel photocatalyst that effectively mimics photosynthesis, in this case $CO_2$ is converted into methane [160]. Another possibility is to use molten carbonate cells to capture $CO_2$ and at the same time, produce electricity [161]. Although all these $CO_2$ sequestration technologies provide radical solutions, they are still immature and possibly cost ineffective. CCS is a potential solution for deep decarbonization, but this technology does not ensure 100% of $CO_2$ emissions are captured [36], especially if embodied energy is taken into account.

## 6. Reduction of Fuel Consumption via Technical and Operational Measures

### 6.1. Vessel Speed

Reducing the average speed of a vessel can reduce the fuel consumption [1,162], however, this has a longer transportation time. Longer transportation time implies more ships or load are required, which reduces the fuel consumption saving [1].

The various vessel speeds can be defined as [163] (and as an example, the relative speeds are given for ocean carriers and shippers):

- Full speed: 23–25 knots (44 km/h).
- Slow steaming: 20–22 knots (39 km/h).
- Extra slow steaming: 17–19 knots (33 km/h).
- Super slow steaming: 15 knots (28 km/h).

Since sea vessels are more fuel efficient at low speeds, "slow steaming has become a widely adopted practice to reduce bunker costs" [162]. There are many emission reduction techniques (such as kite sails discussed in earlier sections), and slow steaming can be "an immediate approach for carriers to improve their environmental impacts" [164,165]. The benefits of slow steaming depend on "ship type, size, routes and duties" [166]. As already mentioned, slow steaming increases transportation times, however, "a 10% reduction in speed may result in a total average emissions reduction of 19%" [167]. Slow steaming can be applied to all ships and sizes and is "considered a technically mature option" [168]. Slow steaming can also help regulate the fleet capacity due to low demand, especially during periods of economic downturn, such as the 2008 financial crisis [163]. On the other hand, slow steaming may cause serious damage to the engine as it operates for longer periods of time at part load conditions [168].

*6.2. Reduction of Hull Resistance*

A smooth hull surface can reduce drag and hence reduce fuel consumption, with a direct impact on $CO_2$ emissions. Reduction of hull resistance can be achieved via vessel design and propeller design.

### 6.2.1. Cleaning

Regular cleaning of the hull's surface can help maintain the ship's designed fuel consumption. The hull's surface contains bacteria that can attract organisms such as seaweed, bivalves and mussels, which increase the drag coefficient [1,169]. Note that the "average surface roughness of a typical ship hull increases by 40 μm/year", which corresponds to an increase in fuel consumption of ~1% [169].

### 6.2.2. Paints and Low-Resistance Coatings

Paints and low-resistance hull coatings can help minimize drag resistance. A "significant capital is invested in anti-fouling paints to prevent bacteria from attaching to the hull" [1,170,171]. These coatings/paints have important anti-corrosion and anti-fouling properties that prevent fouling and offer protection against seawater [172]. This is a mature technology, which has been used for many years [1].

### 6.2.3. Vessel Design

Hydrodynamic optimization is an effective and robust design method that has a key role in the optimization of hulls [173]. Reduction of hull resistance can be achieved via vessel design, such as optimum hull dimensions, reduced ballast operation, lightweight construction, low profile hull openings, interceptor trim plates, skeg shape-trailing edge and bulbous bow.

Reduced Ballast Operation

Lighter displacement means lower wetted hull surface and results in lower resistance. Ballast must be sufficient to preserve stability, handling (e.g., to avoid hull slamming) and immersion of the propeller at optimum depth. Depending on specific design, the maximum fuel consumption reduction is up to ~7% [19].

Lightweight Construction

For new-build ships, light materials, such as aluminum, can be used to manufacture lightweight structures which can help reduce the ship's weight. For existing ships, the weight of the steel structure can be reduced by 5–20%, depending on the amount of high-tensile steel strength. A 20% reduction in weight can reduce the power requirements by up to 9%, however, a 5% reduction is more realistic in most cases [19].

Optimum Hull Dimensions

The hull shape can be designed and optimized, especially in the era of digital twins, to reduce the ship's resistance. Companies are heavily investing in the fourth Industrial Revolution, which includes digital twins. Digital twins help in the design and testing of a new product via 3D-CAD models and real-life models. In this way, R&D costs, manufacturing costs, testing time and hence time for a new product to reach the market are reduced. The concept of digital twins should be applied by the industry and academia in the shipping industry for optimal design decisions, such as hull dimensions. Note that digitalization is also supported by the European Community Shipowners' Association, as stated in their position paper [31].

The optimum length and hull fullness (L/B) ratio can have a big impact on ship resistance. A high L/B can reduce resistance, but a too-high L/B ratio can lead to a large wetted surface, thus increasing resistance. A low L/B implies blunt hull lines, thus increasing resistance. Thus, optimizing and effectively designing new ships taking into account different design parameters can reduce engine demand. However, designing new ships is an expensive option with a long payback period [19].

Low-Profile Hull Openings

Designing low-profile hull openings, thus minimizing the effects of turbulence from bow thruster tunnels, sea chest openings can reduce resistance and thus fuel consumption. Designing and optimizing these openings can achieve up to 5% lower power demand [174].

Interceptor Trim Plates

Interceptor trim plates are a vertical underwater extension at the rear of the hull that direct high-pressure flow behind the propellers downward thus create a lift effect. This is an option that is suitable for relatively high-speed vessels, such as RoRos and ferries. The corresponding fuel efficiency improvement is 1–5% for low power demand and 4% improvement for a typical ferry [174].

Skeg Shape/Trailing Edge

The skeg is an extension of the hull leading up to the propeller shaft line and disk. The skeg's shape is designed in such a way that the flow is evenly directed to the propeller disks, where at low speeds, the flow remains attached, leading to reduced engine power requirements; ultimately, leading up to ~2% fuel efficiency improvements [19].

Bulbous Bow

The wave system generated by the bulb interferes with the wave system of the ship. Hence, a bulbous bow, which is an extension of the bow below the waterline, creates a crest ahead of the ship and thus improves the water flow around the hull [175]. This therefore reduces the drag for large vessels operating within commercial speed ranges with up to 20% lower fuel consumption [19].

6.2.4. Propeller Design

Reduction of hull resistance can be achieved via:

- Wing thrusters (<10% improvement),
- Counter rotating propellers (<12% improvement),
- Optimization of propeller–hull interaction (<4% improvement),
- Propeller–rudder interactions (<4%),
- Advanced propeller blade sections (<2%),
- Propeller tip winglets (<4%), propeller nozzle (<5%),
- Constant vs. variable speed operation (<5%),
- Pulling thruster (<10%), and
- Propeller efficiency management (<2%).

### 6.3. Air Lubrication

Air lubrication is an innovative concept that reduces hull friction using air as a lubricant. "A layer of air is generated between the specially profiled underside and water surface, so that the vessel effectively glides through the water, reducing drag by 5–15%" ([176], p. 20). Fotopoulos and Margaris [177] numerically investigated two separate geometries with a commercial computation fluid dynamics (CFD) software, looking at the effect of air lubrication on fuel consumption, and estimated that it can be reduced by 8%.

There are three methods of air lubrication [177,178]:

1.   Bubble Drag Reduction (BDR),
2.   Air Layer Drag Reduction (ALDR), and
3.   Partial Cavity Drag Reduction (PCDR).

#### 6.3.1. Bubble Drag Reduction

The Mitsubishi Air Lubrication System (MALS), which is a bubble injection system, was first tested and numerically investigated and then applied by Mitsubushi Heavy Industries on a new built ship, the Till-Deymann [179,180]. Mitsubushi Heavy Industries also developed Mitsubishi Turbo-blowers, blowers specifically for MALS to achieve air lubrication [178]. Another example of a lubrication system is the Winged Air Induction Pipe (WAIP), which is "a series of small air chambers fitted with a foil for ultra-fine micro-bubble generation" [178]. Samsung Heavy Industries developed the SAVER system, where it "uses a series of air dispensers installed on the bottom of the ship to spray air bubbles that form an air carpet at the bottom of the ship to reduce frictional drag resistance". [178] Silverstream Technologies have a patented air release system (Silverstream System), where a layer of micro-bubbles is released in the hull to reduce frictional drag resistance [178]. Foreship Air Lubrication System is another real-life example of a bubble drag reduction technique which uses air dispensers positioned on the underside of the hull [178]. Note that the dispensers are hydrodynamically designed, and Foreship claimed that when the system is not in use, the drag resistance does not increase [178].

#### 6.3.2. Air Layer Drag Reduction

When enough air is introduced into the near-wall region of a turbulent boundary layer of water, it will form a continuous layer of air separating the wetted hull surface from the water flow, thus reducing drag reduction [178]. This system has been examined experimentally (such as in References [181–183]) but, to the authors' knowledge, it has not reached commercialization.

#### 6.3.3. Partial Cavity Drag Reduction

In Partial Cavity Drag Reduction (PCDR), a cavity of air is created by a volume of gas with the help of a recess on the bottom of the hull [184]. The cavity between the hull and outer flow reduces the drag resistance. Note that gas is continuously injected into the cavity to replenish some of the lost gas due to entrainment at the cavity closure [184]. The Air Chamber Energy Saving (ACES) System by Damen Ship is an example of a commercial application of PCDR [178].

### 6.4. Waste Heat Recovery

About 50% of the energy is lost due to irreversibilities [185–187] and rejection of heat to satisfy the Second Law of Thermodynamics, however with waste heat recovery (WHR), some of this energy can be recovered from the exhaust gases, which will lead to less emissions and lower fuel consumption [116]. Potential power generation can come from: (1) jacket water (5.2%), (2) air cooler (16.5%) and (3) exhaust gases (25.5%) [188].

With a WHR system, the heat from the exhaust gases is used to drive steam turbines for electricity production for auxiliary power production. Hence, fuel savings come from the auxiliary engines. WHR systems are reasonably applied to ships with high waste heat production and a high consumption of electricity. The literature reports various estimates of

the reduction emission potential; hence, the IMO reports a potential of 8–10% [167,189,190]. Other studies report a fuel saving potential of 4–16% [185,186,191].

There are several technologies available for WHR systems on ships, with a range of efficiencies [186], namely:

- Rankine Cycle (RC):
  - Steam/conventional Rankine Cycle
  - Organic Rankine Cycle (ORC)
  - Super-Critical Rankine Cycle
- Kalina Cycle (KC)
- Exhaust gas turbine system:
  - Hybrid turbocharger
  - Mechanical turbo-compound system
  - Hydraulic turbo-compound system
  - Electric turbo-compound system
- Thermoelectric generation (TEG) systems

Note that the ORC uses an organic fluid as the working fluid [186]. Due to its simplicity, efficiency at low temperatures and moderate costs, it is used for most small-scale WHR systems [192]. On the other hand, the KC uses aqueous ammonia, with different boiling points, as its working fluid, which allows more heat to be extracted [186]. Shu et al. [193] and Singh et al. [186] provide general overviews of different WHR technologies including the TEG, refrigeration, desalination, turbocharging, steam RC, KC and ORC. Mondejar et al. [194] presented a review of ORC for maritime applications. Finally, Zhu et al. [187] comprehensively discuss the highly efficient bottoming power cycles for maritime applications. The main conclusions from Zhu et al. [187] are:

1. There is no optimal WHR solution for maritime applications. One has to consider the trade-offs of costs, working fluid characteristics, size and safety.
2. As a rule of thumb:
   a. Steam RC in combination with a power turbine for marine engines "with power output greater than 25 MW is more likely to be used due to high efficiency and technology maturity".
   b. ORC technologies are recommended for relatively small size ships due to high efficiency and flexibility in recovering waste heat from different sources.
3. The $CO_2$-based power cycles are "more appealing where the system size is of particular importance".
4. KC has found use in land-based applications (such as geothermal applications) but it is seldom recommended in maritime applications because of the toxicity of the working fluid (ammonia-water mixture).

Larsen et al. [195], on the other hand, conclude that the KC cycle has "no apparent advantages" compared to ORC and steam RC. They also mention that even though the ORC cycle "has the greatest potential" for reducing fuel consumption, the working fluids have high global warming potential and hazard levels, which make the technology less attractive. Furthermore, Larsen et al. [195] mention that despite the fact that steam RC is less efficient compared to ORC, it is mature, harmless to the environment and less hazardous.

Zhu et al. [187] also present a techno-economic performance of selected WHR applications for maritime applications. From their survey, they conclude that the payback period of RC and ORC technologies typically lies in the range of 3–8 years, which testifies to the economic feasibilities. They further conclude that the KC- and $CO_2$-based power cycles are "well-established solutions" for recovery of waste heat from biomass combustion exhaust and geothermal sources, however, economic feasibility studies have not been fully conducted for marine applications.

WHR systems have a high technological maturity, commonly used in ships with a "high production of waste heat and a high consumption of electricity" [167]. The IMO [167]

illustrates the costs (in 2007 prices) of WHR systems in different vessel types. In their report IMO [167] suggest that there is a high initial capital cost [1] but fuel savings may result in a payback period between 3 and 10 years depending on fuel prices [196]. However, it should be noted that even though retrofitting WHR on existing ships is a commercially viable option, it may not be possible to do so from a technical viewpoint [196–198]. Theotokatos and Livanos [196] in their study also conclude that from an economic standpoint, a two-stroke engine propulsion plant appears to be the most cost-effective option taking into account discount rates, fuel prices and ship utilization rates. They also mention that a four-stroke engine with WHR could have the same cost-effectiveness as a two-stroke implementation when it exhibits a 22% price increase [196].

## 7. Future Trends, Challenges and Conclusions

This review paper examined the possible pathways and possible technologies available that will help the shipping sector achieve the IMO's deep decarbonization targets by 2100. There has been increased interest from the shipping sector's important stakeholders regarding deep decarbonization. However, deep decarbonization will require financial incentives and policies at an international and regional level given the maritime sector's 3% contribution to GHG emissions [1,6].

Shell and Deloitte conducted a market survey in order to understand the current market trends [4] and it has been realized that the majority of the stakeholders considered decarbonization an important or top priority for their organizations [4]. This, therefore, shows that the market and the industry are considering decarbonization as part of their business strategy. For example, Maersk (the world's largest shipping container company) has announced its intentions to be net-zero carbon by 2050, with carbon-neutral vessels commercially viable by 2030 [199]. The IMO has introduced measures for new and existing ships (SEEMP, EEDI and the fairly recent EEXI), which combine operational and technical measures to help ship operators achieve reductions in emissions.

Section 2 of this review paper discussed alternative fuels that can be used in shipping. The relative advantages and disadvantages of alternative fuels, in terms of cost, technical difficulties and maturity, were presented. Shell's view on future pathways involves a "poly-fuel scenario", in other words, the use of different fuels [54]. Japan's report on "Roadmap to Zero Emission from International Shipping" mentions that pilot concepts are based on two possible emission reduction pathways: (1) LNG, provided that LNG transitions to carbon-recycled methane, and (2) adoption of hydrogen/ammonia as fuel [20]. Although hydrogen/ammonia fuels are a very promising solution to achieve deep decarbonization, there are still issues to be resolved in order for them to be a commercially viable solution. Issues of storage, transportation, safety, toxicity and cost are the prime inhibitors for these alternative fuels. LNG is a very promising solution to achieve short-term decarbonization, with existing ships already deploying LNG (mostly as a drop-in-fuel, blended with existing fuels in marine sector).

Biofuels look marginally more competitive than fuels derived from renewable electricity or from natural gas with carbon capture and storage. However, biofuels have challenges in terms of sustainability and availability, thus, in the mid–long-term, they may be uncompetitive due to sustainability restrictions and price volatility [36]. Because fuel price is the dominant factor that determines the total cost of operation, a fuel derived from natural gas or from a renewable energy source "may offer longer term benefits" compared to biofuels if the future growth in energy global demand and the aforementioned sustainability and availability issues of biofuels are taken into account [36].

Section 3 discussed the various renewable energy sources that are available or currently under development for shipping. The technologies discussed were wind (via the use of kite sails, Fletnner rotors, fixed sails), solar (PV) and biomass. There are existing ships as well as promising technologies that will help the shipping sector reduce the $CO_2$ emissions, however, the technologies are still not mature enough to achieve deep decarbonization on their own.

Section 4 discussed the maturity of technologies currently available that can help the shipping industry achieve deep decarbonization. Marine diesel ICEs are the dominant technology to provide a ship power for propulsion and for its ancillary and auxiliary needs. Given the ICEs dominance in the shipping sector, they are not expected to be replaced any time soon. Hence, it is evident from the research and market trends that the use of combusting blended fuels (e.g., blending hydrogen/ammonia/biofuel with marine diesel fuel) in ICE marine diesel engines is the way forward, at least in the short term. There are existing ships that use hydrogen as a drop-in-fuel, such as Hydroville by CMB. The technology is a hybrid engine that allows it to run on diesel and hydrogen [107]. In addition, the CMB technology is building several hydrogen-powered vessels.

Fuel cells with hydrogen or ammonia as fuels is a promising technology, especially green hydrogen and green ammonia, because not only do they reduce/eliminate GHG emissions, but they also eliminate pollutants ($NO_x$ and $SO_x$). However, fuel cells in shipping applications have lots of issues to resolve, but there are exciting projects underway to test and validate this technology. Projects such as ShipFC, a project funded by Fuel Cells and Hydrogen Joint Undertaking (FCH JU) under the EU's Horizon 2020 research and innovation program, will install a maritime fuel cell that will be powered by green ammonia [147]. The use of hydrogen in fuel cells has attracted lots of commercial interest. Recently, a Danish–Norwegian project aimed to build and test a ferry, the Europa Seaways operated by DFDS ferries, powered by a hydrogen fuel cell. The project recently gained EU funding and aims to have Europa Seaways operational by 2027. Note that this project attracted several key players in the shipping and energy sectors, who joined forces to build a ferry that will be able to transport 1800 passengers [145].

Section 5 discussed the various $CO_2$ abatement options, which in essence discussed strategies and techniques that can help reduce fuel consumption, such as vessel speed, vessel and propeller design, waste heat recovery and even cleaning and coating of the hull's surface. These techniques reduce fuel consumption by either reducing the ship's resistance, improving propulsion or utilizing the wasted heat energy from the ship's engine for auxiliary and ancillary power needs. With WHR technologies, the most promising concepts are to reduce fuel consumption and thus $CO_2$ emissions. Steam and organic Rankine Cycles seem to be the most promising technologies, as they are mature. Other technologies, such as the Kalina Cycle, require further research, especially in terms of techno-economic feasibility if this technology is to compete with the more mature steam and organic Rankine Cycles. Note that the techniques discussed in Section 5 are short-term measures to achieve reduction of $CO_2$ emissions, but will not achieve complete decarbonization on their own because the main source of energy is fossil-based fuels.

The IMO's targets will be achieved via a radical technology shift together with the aid of social pressure, financial incentives and regulatory and legislative reforms at the local, regional and international levels.

**Author Contributions:** Conceptualization, G.M. and E.A.Y.; methodology, G.M.; formal analysis, G.M. and E.A.Y.; investigation, G.M.; resources, G.M.; data curation, G.M.; writing—original draft preparation, G.M.; writing—review and editing, G.M.; visualization, G.M.; supervision, E.A.Y.; project administration, G.M.; funding acquisition, E.A.Y. All authors have read and agreed to the published version of the manuscript.

**Funding:** This research was funded by CMMI Cyprus Marine and Maritime Institute. CMMI was established by the CMMI/MaRITeC-X project as a "Center of Excellence in Marine and Maritime Research, Innovation and Technology Development" and has received funding from the European Union's Horizon 2020 research and innovation program under grant agreement No. 857586 and matching funding from the Government of the Republic of Cyprus.

**Institutional Review Board Statement:** Not applicable.

**Informed Consent Statement:** Not applicable.

**Data Availability Statement:** Not applicable.

**Conflicts of Interest:** The authors declare no conflict of interest. The funders had no role in the design of the study; in the collection, analyses, or interpretation of data; in the writing of the manuscript, or in the decision to publish the results.

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
