# Peer review of "Decarbonization in Shipping Industry: A Review of Research, Technology Development, and Innovation Proposals"

_jmse, doi:10.3390/jmse9040415_

Round 1

Reviewer 1 Report

This is a review paper which deals with a very interesting and hot topic: Decarbonization of shipping industry. The subject is very challenging and since research is still on-going it is inherently difficult to cover completely the whole field in a review. However, since this is the scope of the paper (a review), it is advised and this would greatly improve the impact of the study, to critically present the various techniques proposed by the industry and research community, and not just refer to the main conclusions of each study. In a review, the authors have the opportunity to interpret in a transparent and technically sound way all the information and data collected from the various sources, in order to help readers to gain a global and solid perspective of the subject. This is the main weakness of the present work, and it is strongly suggested to add this feature in the revised version since most of the tough work has already been done (collection of data) and will greatly improve the quality and impact of the final work.

Specific comments/remarks have been added as annotations in the attached document.  It is suggested that the authors should take into account these comments and submit a revised version of the paper.

Reviewer 2 Report

.See the attached Word file, containing comments for both Editors and Authors. Based on the overall quality of this manuscript I would reject it, but since the topic is important and the authors aim to bring together insights from a wide range of sources I have also added some comments for a major revision, in case the other reviewer/s make such recommendation or the authors end up revising and submitting elsewhere. With the wide reading that has been done, it would be nice if the authors could develop this into a proper review article.    

Reviewer 3 Report

The work is focused on decarbonization in the marine transport sector to meet the IMO 2050 targets.
The document is not a scientific work that reports analysis of experiments but is more a review report. The work is well written and discusses in detail all the strategies to reduce carbon emissions, also reporting future fuel alternatives with zero impact.
The work is very detailed and very useful as a guideline in the maritime transport problem, it also meets the aims of JMSE and should be published in my opinion.
However, I suggest to the authors to pay attention to the format of the references used in the text, there are some errors. Also, the document should be checked to improve the English.

Round 2

Reviewer 2 Report

I see that the other reviewers find the work acceptable, and also note that several of my comments were followed up. My overall rating of the paper remains more or less the same, since there has not been any major revision (e.g. I don't see any real description of review methodology). However, as I also noted first time, I think many will be interested in reading this anyway, as a fresh summary of recent reports and and studies on a highly important topic, and in my experience it is also well written. Therefore I now support acceptance.